# Distinct causes underlie double-peaked trilobite morphological disparity in cephalic shape
Harriet B. Drage [1] ✉ & Stephen Pates [2,3]

Trilobite cephalic shape disparity varied through geological time and was integral to their ecological niche diversity, and so is widely used for taxonomic assignments. To fully appreciate trilobite cephalic evolution, we must understand how this disparity varies and the factors responsible. We explore trilobite cephalic disparity using a dataset of 983 cephalon outlines of *c.* 520 species, analysing the associations between cephalic morphometry and taxonomic assignment and geological Period. Elliptical Fourier transformation visualised as a Principal Components Analysis suggests significant differences in morphospace occupation and in disparity measures between the groups. Cephalic shape disparity peaks in the Ordovician and Devonian. The Cambrian–Ordovician expansion of morphospace occupation reflects radiations to new niches, with all trilobite orders established by the late Ordovician. In comparison, the Silurian–Devonian expansion seems solely a result of within-niche diversification. Linear Discriminant Analyses cross-validation, average cephalon shapes, and centroid distances demonstrate that, except for Harpida and the Cambrian and Ordovician Periods, order and geological Period cannot be robustly predicted for an unknown trilobite. Further, k-means clustering analyses suggest the total dataset naturally subdivides into only seven groups that do not correspond with taxonomy, though k-means clusters do decrease in number through the Palaeozoic, aligning with findings of decreasing disparity.

## Analysing morphological disparity in the fossil record

The fossil record documents the origins and radiations of animal groups and their morphological variation. Critically, the fossil record provides a perspective of how life on Earth has changed through time, including abundant information on long-extinct groups, allowing inferences about the impacts of environmental change, extinction events, and radiations on the diversity and morphological disparity of animal groups. This morphological disparity, which is often decoupled from taxonomic diversity[1–6], results from interlinked ecological, functional, and taxonomic components[7–9]. Disparity is often quantified through morphometry and visualised as morphospaces, with relative trends in morphospace occupation area and location through time for clades informing on the patterns and processes that impacted their morphological variation on geological timescales. Increases or decreases in the volume of morphospace occupied (often quantified through metrics such as sums of ranges, and sums of variances[10,11]) indicate a corresponding increase or decrease in the morphological disparity of a group, often caused by extinction or radiation events lacking selectivity or acting only at the margins of the morphospace[12]. A complementary metric, the change in

position of the centroid in morphospace through time (or average position in the morphospace), indicates a change in the mean shape of members of the group, which can be interpreted as selective extinctions and radiations favouring particular morphologies, likely linked to an ecological change or environmental factor[12].

Trilobites are one of the most abundant animal groups preserved in the fossil record, owing to their notable global diversity across the Palaeozoic, their strongly biomineralised exoskeletons with high preservation potentials, and their colonisation of a range of marine environments worldwide[13–18] (Fig. 1). Thus, they provide an excellent study group with which to explore the interrelated temporal (including changing ecological and functional pressures) and taxonomic components acting on the morphological disparity of a diverse clade. Trilobites originated during the early Cambrian, with the oldest body fossils of this group *c.* 521 million years old[19]. Analyses of their raw diversity have revealed two major peaks in the global record, the largest in the middle Cambrian[20] and the second in the early Devonian[21,22], before a dramatic reduction during the late Devonian[23,24], leaving only members of the order Proetida (plus Aulacopleurida if

¹Institute of Earth Sciences, University of Lausanne, Lausanne, Switzerland. ²Department of Zoology, University of Cambridge, Cambridge, UK. ³Centre for Ecology & Conservation, University of Exeter, Penryn, Cornwall, UK. ✉e-mail: harriet.drage@unil.ch

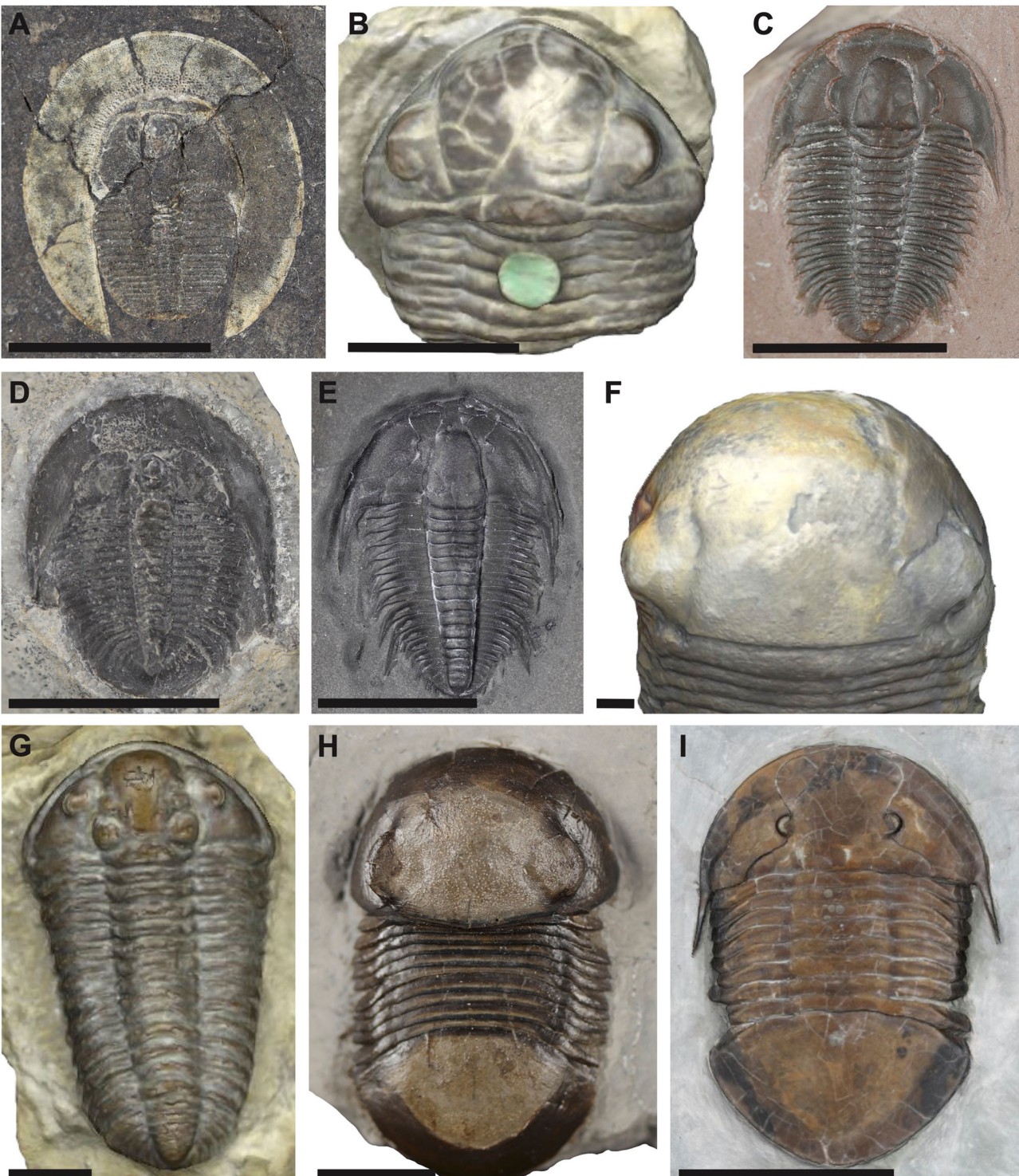

**Fig. 1 | Examples of trilobite specimens included within the dataset.**
**A** *Bohemoharpes naumanni* Senckenberg Museum X286a; **B** snapshot of 3D model, *Acaste downingiae* NHMUK 44409; **C** *Modocia typicalis* Senckenberg Museum unnumbered Westphal collection; **D** *Pterocephalia norfordi* Senckenberg Museum unnumbered Westphal collection; **E** *Orygmaspis contracta* Senckenberg Museum unnumbered Westphal collection; **F** snapshot of 3D model, *Bumastus barriensis* NHMUK I1029; **G** snapshot of 3D model, *Calymene blumenbachii* NHMUK 44215; **H** *Bumastus iorus* Senckenberg Museum unnumbered Westphal collection; **I** *Megalaspides* sp. SMNKPAL 3887. Scale bars = 1 cm.

recognised as an order). Species diversity reduced further by the Carboniferous and trilobites finally went extinct at the end Permian 252 Ma[23]. While the middle Cambrian peak in diversity correlates with a peak in intraspecific variation for the group[20], previous studies have differed on when the peak in trilobite disparity occurred. Earlier studies assessing changing disparity over geological timescales focused on the outline shape of the cranidium and found a disparity peak in the Ordovician[25,26], facilitated by environmental preferences of taxa[8]. However, a more recent study using the outline of the whole cephalon (cranidium and librigenae), albeit with a considerably smaller sample size (400 compared to 1125 in Foote[26]), found an earlier Cambrian peak in disparity and a second peak in the Devonian, broadly mirroring reported diversity trends[27].

While morphometric analyses on trilobite data have been used to address a range of macroevolutionary[1,8,25–28], behavioural[27,29,30], developmental[31–33], systematic[34–37], and taphonomic[33,38] questions, several crucial macroevolutionary questions remain.

1. What are the extents of trilobite disparity in cephalic morphometry?
2. How does cephalic morphometry vary with order-level taxonomy, and can cephalon shape be used to predict taxonomic assignment?
3. How does cephalic morphometry vary across the Palaeozoic, in terms of morphospace volume occupation and movement through this space, and can cephalon shape be used to predict the geological Period occupied?

To address these questions, we analyse a dataset of nearly 1000 trilobite cephala, representing over 500 species, digitised as 2D outlines. We quantify the taxonomic (ordinal level) and temporal changes in morphospace occupation for Trilobita, documenting how the morphology of this key Palaeozoic group changed over the course of 250 million years. We also provide all data and R code (https://osf.io/vz9a5/)[39], and the analytical protocol for these analyses.

## Previous works on the disparity of Trilobita or groups therein

There is an extremely rich history of both traditional and geometric morphometric approaches aimed at furthering our understanding of varied aspects of trilobite evolution. In fact, the wealth of these studies as applied to specific trilobite species or groups are too numerous to fully cover outside of a dedicated review[7,25,26,40–59]. The use of elliptical Fourier analyses to quantify the disparity of trilobite cranidia was pioneered by Foote[28]. This dataset was then expanded[1,25,26] and used to explore links between environmental and morphological characteristics within the group[8]. More recently, studies have used morphometric data pertaining to the whole cephalon. A number of key recent studies following these are discussed, though as these examples and the references therein demonstrate, a full review of trilobite morphometric analyses is beyond the scope possible here.

Recent works have quantified trilobite cephalic morphometry using various protocols and large-scale datasets to explore their diversity and disparity through geological time, and their potential links to other characteristics. Suárez and Esteve[27] used 400 cephalon outlines (including agnostids) to assess trends in cephalic morphometry with the prevalence of different enrolment strategies through the Palaeozoic. Bault et al.[24] used a large dataset of cephala and pygidia across trilobite groups to determine whether disparity trends followed diversity across the Devonian. Similarly, Serra et al.[60] presented a Devonian trilobite dataset, landmarking protocol, and morphometric analysis for the cephalic and pygidial outline and some internal structures. Holmes[61] assessed cephalic disparity during the Cambrian, with different cephalic structures suggesting differing patterns in morphospace occupation expansion. Cephalic outline showed rapid diversification in form, while interior structures showed a more gradual increase in disparity[61]. Hopkins et al.[62] analysed diversity, disparity, and geological range in Permian trilobites and compared this to the other Palaeozoic Periods.

Many other studies have used morphometric analyses to explore the diversity and disparity of specific trilobite groups. These analyses have been used to assess shape change during ontogeny, such as for *Cryptolithus tesselatus*[33] Green, 1832, to better understand development trajectories. Several studies explored shape evolution within specific clades; for example, Hopkins[63] coupled cranidial shape change with exoskeleton morphological characters in pterocephaliids, and Bault et al.[64] explored shape evolution in Phacopidae and the associations between disparity and external forcing events. Morphometric analyses have also been much used to aid in taxonomic work[59,65]. These analyses can also be informative for building or testing phylogenetic hypotheses; for example, Martin et al.[37] produced a phylomorphospace for asteropyginid glabellae. Still more research used trilobite morphometric data to explore specific morphological or behavioural evolutionary questions. For example, Vargas-Parra and Hopkins[66] tested patterns of modularity in the trilobite cephalon, and hypotheses

relating to the developmental origins of the eye. Drage[29] used traditional multivariate morphometric analyses to test for an association between body proportions and exoskeleton moulting behaviour across Trilobita, while Drage et al.[30] tested the same hypothesis on an intraspecific scale, using a large dataset of *Estaingia bilobata* Pocock, 1964.

## Results

### Taxonomic variation in cephalon morphometry

The PCA morphospace with data grouped by taxonomic order (Fig. 2) shows that, for the most part, trilobite specimens occupy a constrained region of morphospace, with some general differences between order groups and several interesting deviations. Most orders overlap extensively at the centre of the morphospace (Fig. 2A), demonstrating that most trilobite cephala, no matter their taxonomic assignment, display similar morphologies at least in terms of their common outline shapes. However, the orders show differing morphospace occupation in terms of their extremes, and several groups, most notably the Harpida, are offset from the centre of morphospace and display more limited overlap with the other orders. Asaphida cluster tightly within the centre of morphospace (Fig. 2B) as 'typical trilobites', with most aulacopleurids clustering also within this region at low values of PC1 and 2. Corynexochida cluster to the left of the centre of morphospace occupation at more negative PC1 values, in a similar region to a dense cluster of Phacopida, though the Phacopida also show dense clustering closer to the morphospace centre (Fig. 2B). Though its convex hull is largely encompassed with the phacopid hull, Proetida show clustering within their broader convex hull in an opposing area of morphospace to clusters of the Phacopida, with the majority of specimens falling between the two major phacopid clusters and the rest found along more negative PC2 values. It is evident that the Harpida are particularly divergent in cephalic outline morphometry, with the group forming an almost entirely independent cluster in the most negative region of PC2, though it is interesting that the superficially similar Trinucleida are quite broadly scattered across the morphospace (Fig. 2B). Lichida, Odontopleurida and Olenida occupy similar areas of morphospace to each other, mostly at positive PC2 values and reasonably centred along PC1, while the Redlichiida cluster at slightly more positive PC1 values (Fig. 2B). The LDA morphospace shows comparable results to the PCA (Fig. 2C), with most order groups greatly overlapping, the Harpida highly divergent from this general clustering, and some differences in the extremes of various order groupings. This is reinforced by the LDA cross-validation table (Fig. 3), which confirms the Harpida cluster is notably different to the other orders and demonstrates that the extensive overlap of Phacopida with the other orders would likely cause false predictions for new data points for almost all orders (except Harpida and perhaps Redlichiida). Despite the differences in clustering for Proetida and Phacopida, the overlap of convex hulls means that a new proetid specimen is also more likely to be predicted as a phacopid (Fig. 3).

Pairwise comparisons of the mean cephalon shapes for each order make clear the shape changes responsible for the clustering differences between these groupings (Fig. 4). Harpida shows a highly divergent average cephalon shape, with a much longer (sag.) cephalon with broad and long genal spines. The mean trinucleid cephalon is similar to that of the harpid, but slightly shorter (sag.) and with the genal spines narrower and more distally directed. In contrast, the mean Lichida and Odontopleurida cephala are shorter (sag.) compared to all other orders. The mean Aulacopleurida, Olenida and Redlichiida cephala are almost identical, as are the Corynexochida and Phacopida cephala (with the Asaphida, the unassigned group and Proetida also alike these; Fig. 4). We can therefore divide the orders into five major trilobite cephalon morphogroups: (1) Harpida; (2) Trinucleida; (3) Lichida–Odontopleurida; (4) Aulacopleurida–Olenida–Redlichiida; (5) Asaphida–Corynexochida–Phacopida–Proetida. These morphogroups are quantitatively supported by the centroid distances between the orders when grouped in PCA morphospace (Fig. 4). The Harpida have high centroid distances to all other groups, and the other morphogroups are reflected in the short pairwise distances between their constituent order centroids. However, the centroid distances between the Trinucleida and the other

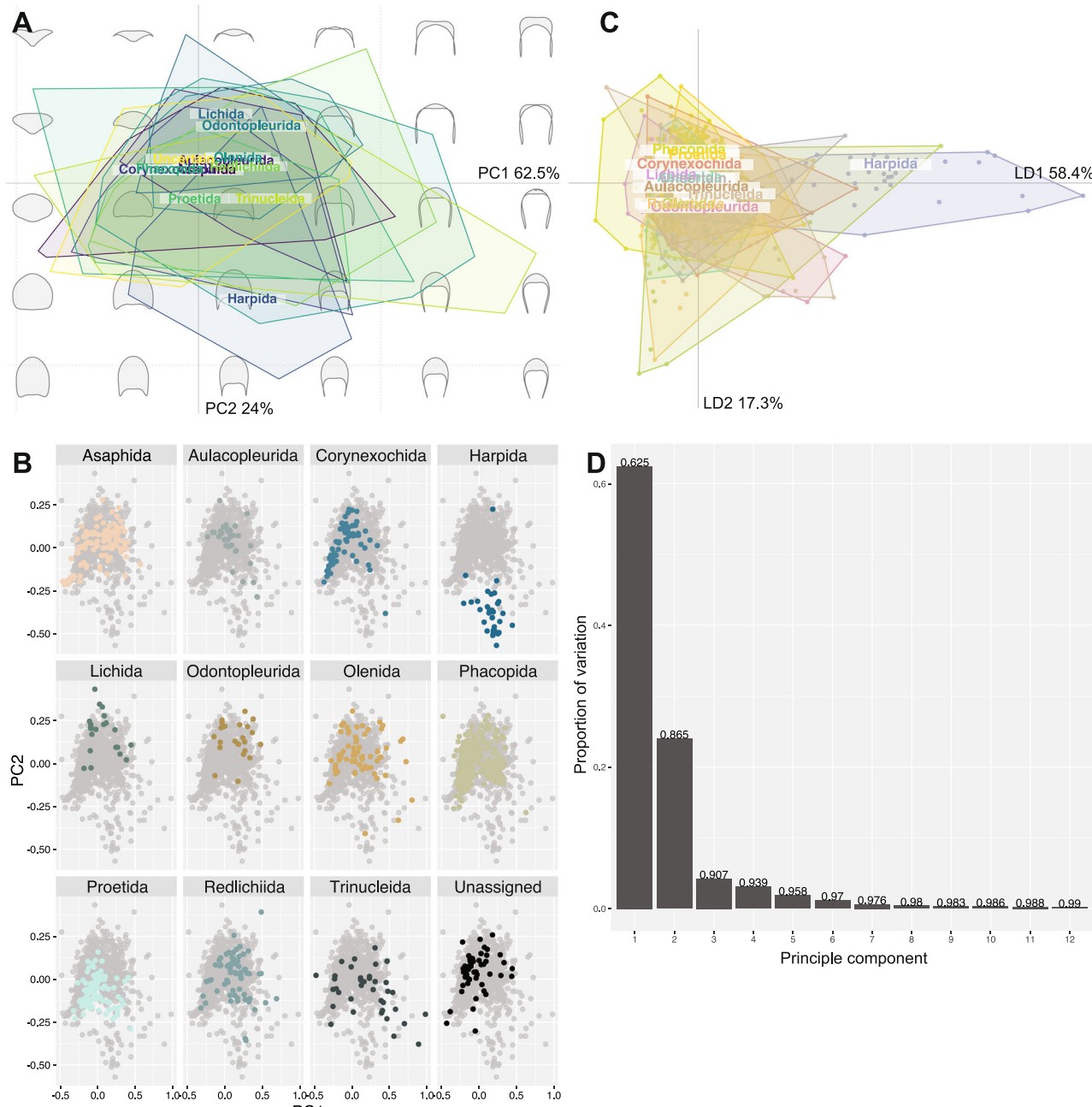

**Fig. 2 | Principal components analysis (PCA) of trilobite cephalic outline morphometry, grouped by taxonomic order. A** PCA total plot; **B** PCA facet wrap plot; **C** linear discriminants analysis (LDA) plot. Principal components scree plot (**D**) is applicable to the entire dataset and contains no grouping. **A**, **B** visualise the same data, but (**B**) has the overlapping convex hull polygons separated out onto individual morphospaces for ease of visualisation. Group labels in (**A**, **C**) display the centroid positions in morphospace. $N = 983$ independent biological samples.

groups are not notably high, and the distance between the Trinucleida and Redlichiida is reasonably low (Fig. 4), suggesting similarity between the average cephalic outlines of these two groups.

A MANOVA test indicates a significant difference between the distributions of PC scores for the order groupings, suggesting that taxonomy influences cephalic outline data ($F = 20.329$, $p = <2.2 \times 10^{-16}$). This is reinforced by many significant pairwise MANOVA tests, testing the null hypothesis that there are no differences in PC score distributions for each pair of orders (see Supplementary Data III for all test results). Of particular interest are the high $F$ values (all $F = > 15$ and significant $p$ values at $p = <7.576 \times 10^{-4}$) for the following pairings, which suggest extensive differences in morphospace distributions: the Harpida with all orders; Lichida–Proetida pairing; Odontopleurida–Phacopida; Odontopleurida–Proetida; Olenida–Phacopida;

and Phacopida–Redlichiida. These results further support the extensive difference of the Harpida from the other orders, and the existence of the general order morphogroups noted above, particularly the Lichida–Odontopleurida grouping as distinct from the other morphogroups.

The Olenida, Phacopida, Redlichiida and Trinucleida occupy the greatest areas of PCA morphospace (Supplementary Table 1; 0.55, 0.48, 0.44, and 0.51 areas respectively), with the other orders being comparably constrained (Fig. 2). The Asaphida, Corynexochida, Harpida, Proetida and unassigned group occupy about a half to two-thirds of the morphospace area of the aforementioned groups (0.31, 0.29, 0.28, 0.25, and 0.31, respectively). The Aulacopleurida, Lichida, and Odontopleurida occupy much less than half of the morphospace area of the groups with the largest areas (0.19, 0.16, and 0.18, respectively). Disparity measures generally

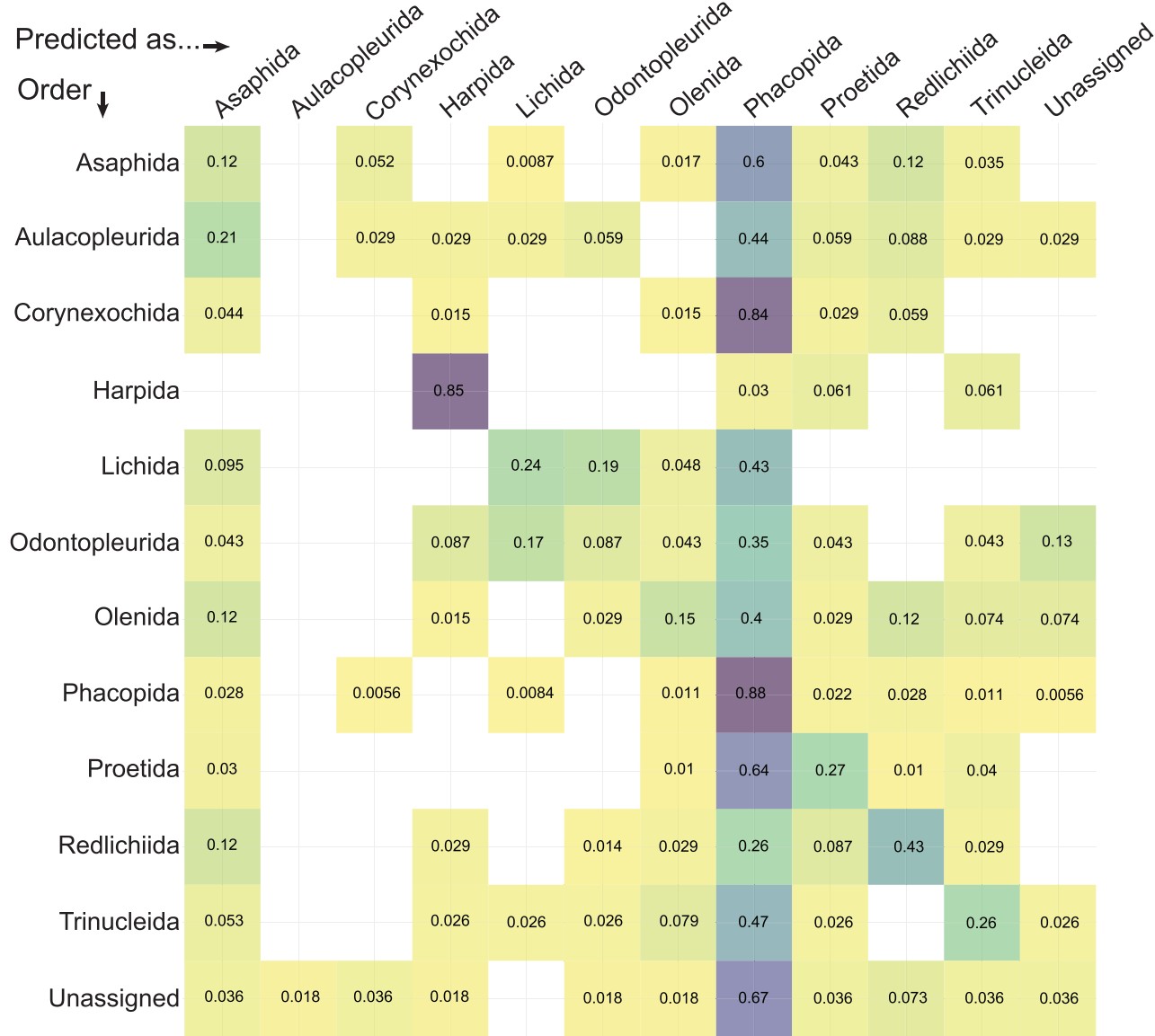

**Fig. 3 | Linear discriminants analysis (LDA) cross-validation table for data grouped by taxonomic order.** *Y*-axis is the true order of a new data entry, and the *x*-axis is the order that this new entry would be predicted as being under this dataset, with a given probability in the relevant cell. Empty cells mean the probability is effectively zero. *N* = 983 independent biological samples.

corroborate this, with Asaphida, Olenida, Redlichiida, and Trinucleida showing high sums of variances (SoVs; Supplementary Table 1; 0.053−0.133), and the SoVs comparably lower for all other orders (Fig. 5A; <0.05). The sums of ranges (SoRs) show similar patterns (Fig. 5B; Supplementary Table 1), with Asaphida, Olenida, Redlichiida, and Trinucleida (1.534−2.023) having high values as for the other measures. However, other orders also show high SoRs, most notably Phacopida (1.674). The SoRs are low for Aulacopleurida, Lichida, Odontopleurida and Proetida (<1.220), though these SoRs are more liable to be impacted by outliers than the SoVs. Together, these disparity measures suggest high within-group disparity for the Asaphida, Olenida, Redlichiida, and Trinucleida, and comparably low within-group disparity for the Lichida and Odontopleurida in particular. The Phacopida interestingly show a comparably large convex hull area and corresponding high SoR, but a relatively low SoV, suggesting lower disparity within-group than indicated by their morphospace area occupation (clustering rather than an even spread across the occupied morphospace area). The Asaphida show high disparity metrics for their medium convex hull area, suggesting reasonably high within-group disparity. Pairwise *t*-tests

support these findings (see Supplementary Data III for all test results), suggesting most order groups vary significantly in both disparity measures (Fig. 5). For example, all pairings with the Trinucleida are strongly significant, supporting a statistically higher disparity in this group.

**Cephalon morphometric variation through geological time**
When grouped by geological Period, convex hulls of trilobite cephalic outlines also show considerable overlap in the PCA morphospace (Fig. 6A), though clear differences in the position and relative areas of morphospace trilobites occupied during each Period are evident. Trilobites occupied larger areas of morphospace during the Cambrian to Devonian (area > 0.40), with a significant constriction in the Carboniferous and Permian (Fig. 6A, B; Supplementary Table 2; 0.14 and 0.096, respectively). This constriction occurs towards the centre of morphospace (and slightly negative on PC2). The PCA morphospace occupancy during the Cambrian is quite diffuse, and only during the Cambrian and Ordovician are the positive extremes of PC1 occupied (Fig. 6B). The Silurian contains some trilobites with negative PC2 values close to the extremes. In comparison, the Devonian shows

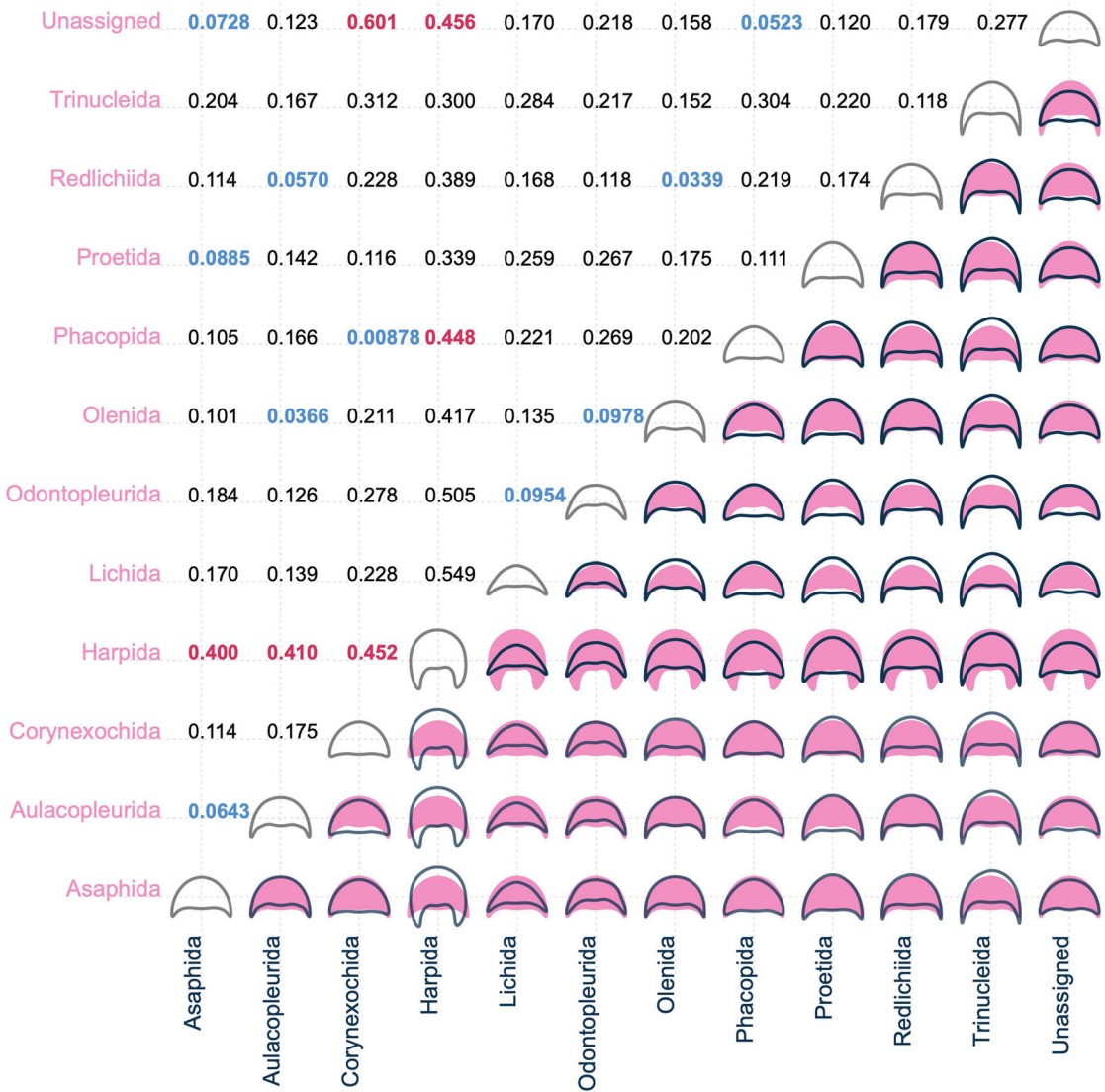

**Fig. 4 | Mean cephalic shape for each taxonomic order compared pairwise, and pairwise centroid distances for the same.** Bottom-right half of matrix: mean cephalic shape for data grouped within each taxonomic order, compared in a pairwise fashion with the colour of the outline reflecting the colour of its axis. Top-left half of matrix: pairwise centroid distances for taxonomic orders, showing their relative positions in PCA morphospace; blue figures indicate particularly close pairwise distances (<0.100), and red figures indicate particularly far pairwise distances (>0.399). $N = 983$ independent biological samples.

greater morphospace occupancy to the extremes of PC2, with occupancy at PC1 extremes already lost by this time. Interestingly, PCA morphospace occupation during the Devonian appears to comprise three mostly distinct clusters (Fig. 6B); one each to the left and right of the centre of morphospace and extended along PC2, and one cluster at the negative extreme of PC2.

The LDA morphospace (Fig. 6C) differentiates the Cambrian and Ordovician at their extremes, while also reinforcing the centroid of the Cambrian as offset to the other Periods. The centroids of the Carboniferous and Permian are close together but separated from the remaining Palaeozoic Periods. The LDA cross-validation table (Fig. 6D) demonstrates the impact of the extensive overlap of all Periods in morphospace with the particularly broad Ordovician convex hull; data from all Periods except the Cambrian are highly likely to be placed within the Ordovician if a new corresponding data point were added. New data points from the Cambrian are more likely to be predicted correctly than placed in a different Period.

The mean cephalon shapes appear reasonably different for each of the geological Periods (Fig. 7). The Permian mean cephalon is notably long (sag.) with short genal spines, while the Cambrian mean cephalon is much shorter (sag.) with longer and narrower genal spines than for the other Periods. Only the mean cephala for the Ordovician and Silurian are similar,

and to a lesser extent the Devonian and Carboniferous; consequently, morphogroups are harder to identify than for the order groupings. However, the pairwise centroid distances are generally low, particularly when compared to those for the order groupings, with the Ordovician–Silurian–Devonian seemingly forming a loose morphogroup with very proximal centroids, and the Carboniferous and Devonian also having centroids close together. The centroid distance between the Cambrian and Permian is notably high (Fig. 7).

Geological age impacted cephalon outline shape, as demonstrated by the significantly different distributions of PC scores for the geological Periods identified by a MANOVA test ($F = 9.966$, $p = <2.2 \times 10^{-16}$). Almost every pairing of geological Periods is significantly different to each other following pairwise $t$-tests, except the Ordovician–Silurian and Carboniferous–Permian (see full test results in Supplementary Data III).

Disparity within the Carboniferous and Permian is low (Supplementary Table 2; Fig. 8; SoV = 0.029 and 0.021, respectively, SoR = 0.995 and 0.831, respectively). The Silurian appears moderately disparate based on its relatively high SoV and SoR (0.047 and 1.556 respectively), while disparity is high in all other Periods (Supplementary Table 2; Fig. 8). The Devonian is particularly disparate based on its SoV (0.078; presumably from its clusters

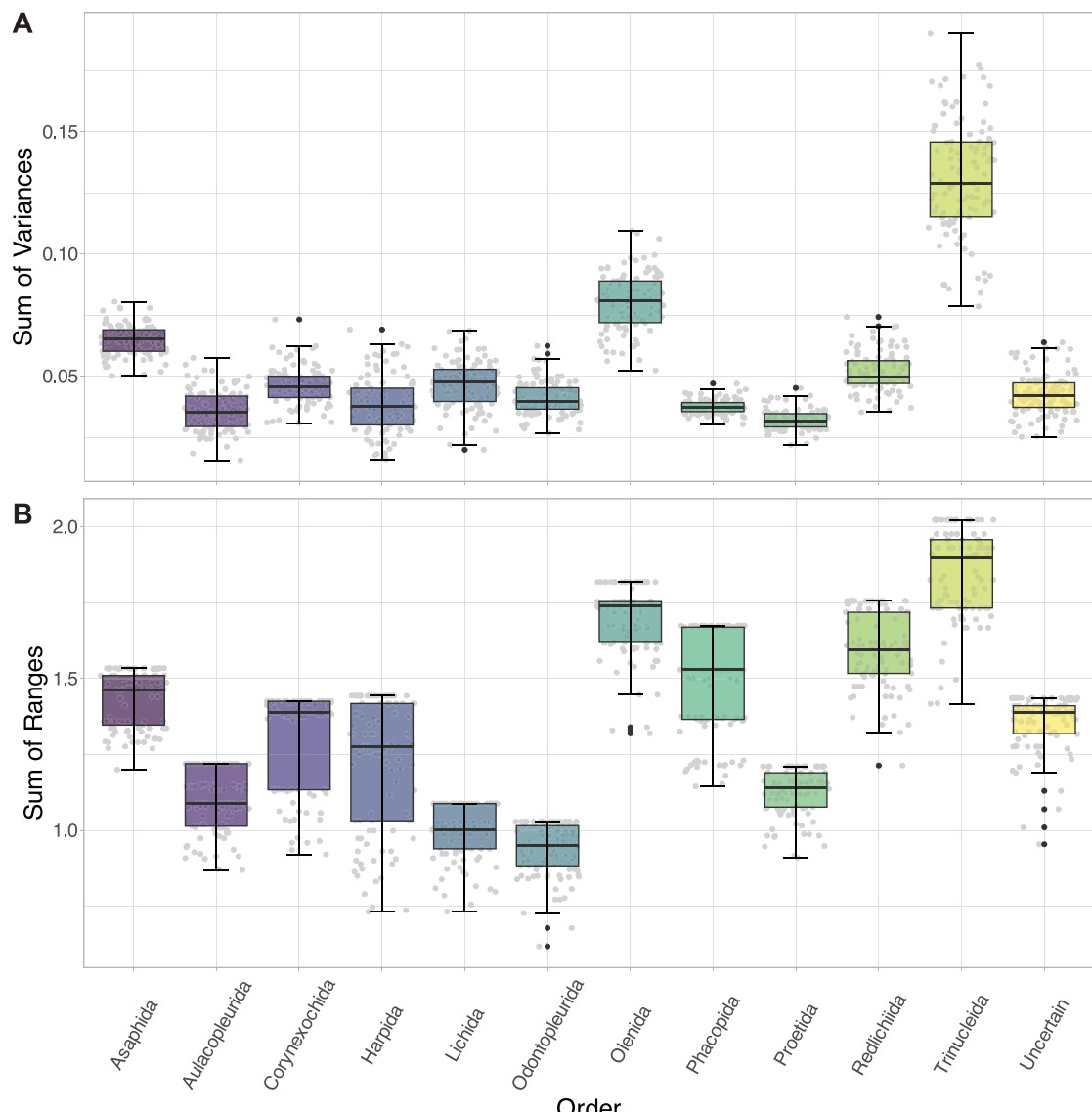

**Fig. 5 | Box and whisker plots showing disparity measures calculated for each taxonomic order grouping. A** sum of variances; **B** sum of ranges. The black lines show the median value, the hinges correspond to the first and third quartiles, and the error bars represent 1.5 × the interquartile range. Black plotted points represent potential outliers (outside of the error bars), and grey plotted points show the total underlying data. $N = 983$ independent biological samples.

being very distant in morphospace), and the Ordovician based on its SoR (2.217; high morphospace occupation). All pairwise *t*-tests were significant, demonstrating that the geological Periods differ in their relative disparities (see Supplementary Data III for all test results).

**Changing orders through time**

Presence and area of morphospace occupation clearly differ for the orders through the Palaeozoic (Fig. 9). There is a radiation indicated by the increased number of distinct orders between the Cambrian (7 orders) and Ordovician (11 orders), then extinctions as is clear by a decrease in orders to the Silurian and Devonian (7 orders, for both Periods), followed by a severe extinction into the Carboniferous (2 orders) and Permian (1 order remaining). The large area of morphospace occupied in the Ordovician appears partially a result of the evolution of Trinucleida, with their high disparity measures (Fig. 9B). The Harpida occupy a broad range of values along PC2 in the Ordovician (Fig. 9B), which is reduced to a very extreme position at positive PC2 values in the Silurian and Devonian (Fig. 9C, D). Only the Proetida and Aulacopleurida survive to the Carboniferous (Fig. 9E), with the Aulacopleurida also lost by the Permian (Fig. 9F). From

their origination in the Ordovician to their demise at the end-Permian, the Proetida are consistently positioned close to the centre of trilobite morphospace occupation.

**Natural clustering in the dataset**

K-means clustering analyses produced two major hypothetical clustering topologies, which both subdivide the occupied PCA morphospace into seven clusters (Supplementary Fig. 1A). The modelled clusters at the centre, top, and bottom right of morphospace have similar average cephalic morphometries between the two topologies. However, the two topologies differently split the occupation at the left side of the morphospace; one topology splits this side into three clusters (Supplementary Fig. 1B), and the other into two large clusters (Supplementary Fig. 1C), with the latter instead splitting the right side of the morphospace into three small, densely packed clusters rather than two.

K-means clustering analyses individually carried out during each geological Period demonstrate that stochastic clustering of cephalon morphometry changes across the Palaeozoic (Fig. 10). There is a shift in the dataset from a higher optimal number of clusters early in the Palaeozoic

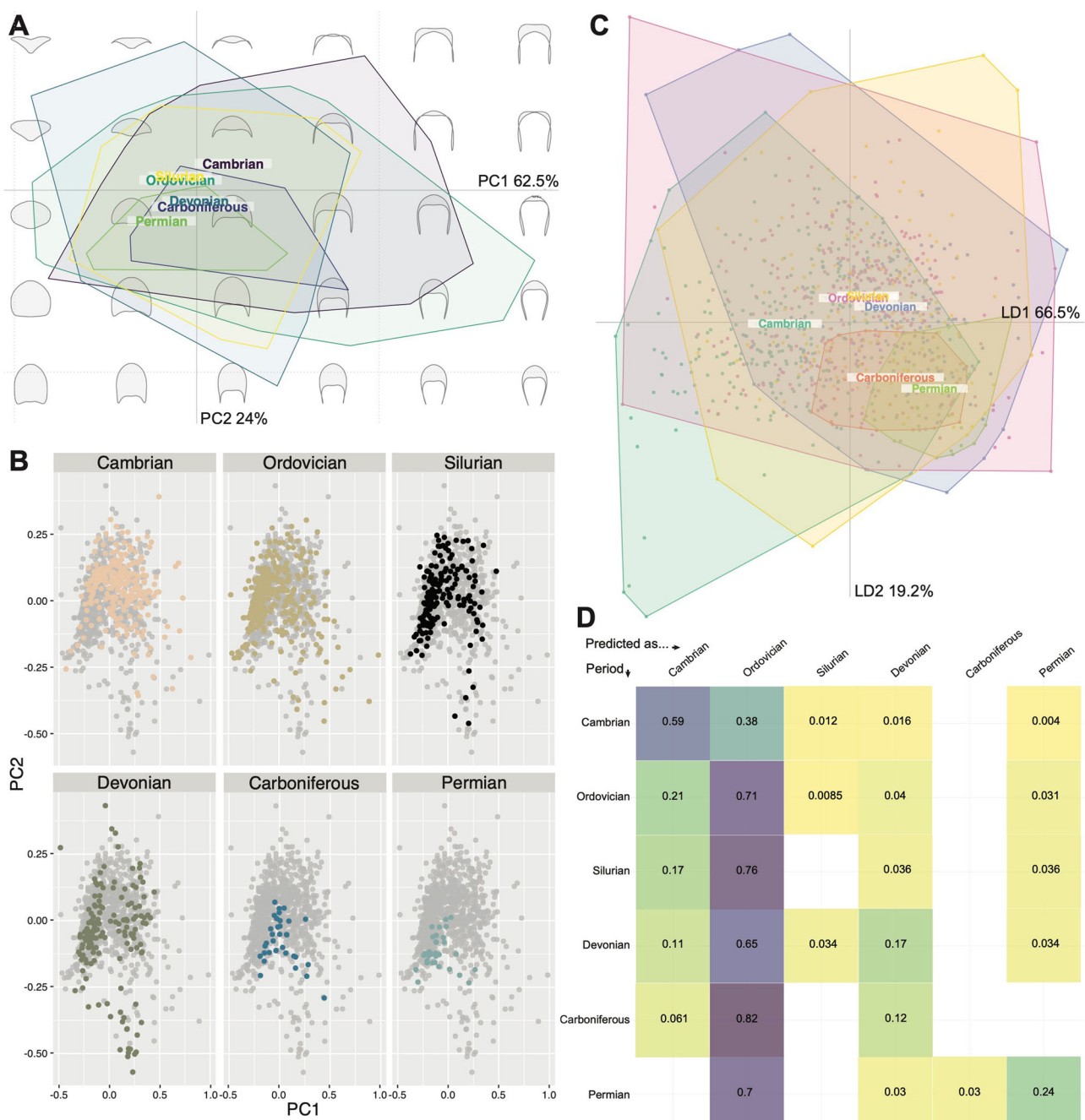

**Fig. 6 | Principal components analysis (PCA) of trilobite cephalic outline morphometry, grouped by geological Period. A** PCA total plot; **B** PCA facet wrap plot; **C** linear discriminants analysis (LDA) plot; **D** LDA cross-validation table.
**A, B** visualise the same data, but (**B**) has the overlapping convex hull polygons separated out onto individual morphospaces for ease of visualisation. For (**D**), the *y*-axis is the true occupied Period of a new data entry, and the *x*-axis is the Period that this new entry would be predicted as being found in under this dataset, with a given probability in the relevant cell. Empty cells mean the probability is effectively zero. The scree plot for the PCA (**A, B**) is the same as that presented in Fig. 2D. Group labels in (**A, C**) display the centroid positions in morphospace. *N* = 983 independent biological samples.

(seven clusters; Fig. 10A–C) to a lower number (five clusters; Fig. 10D–F) from the Devonian onwards. These analyses clearly show an overall cephalic shape trend across the Palaeozoic from a higher disparity of forms earlier, with many axially short cephala with long genal spines, to less disparate cephalic forms that are more axially long and few with long genal spines. This latter is particularly the case in the Carboniferous and Permian (Fig. 10E, F), where the dataset has only clusters of forms below the PC1 axis in morphospace. The clusters present are similar between the Cambrian and Ordovician k-means analyses (Fig. 10A, B), with slightly more subdivision of the axially long cephala with short genal spines than during the post-Ordovician. The clusters on the left of the morphospace (negative PC1) move more centrally (i.e., away from the extreme left) from the Cambrian to the Silurian, while the clusters in the top and right (positive PC1 and 2) remain approximately consistent into the Silurian. However, by the Silurian the central-most cluster has entirely disappeared, with a new cluster appearing at the bottom extreme of the occupied morphospace (negative PC1; Fig. 10C); this new cluster persists into the Devonian but disappears after this time (Fig. 10D). During the Devonian, several clusters are again positioned centrally on the PC1 axis, with the Carboniferous and Permian both also having a cluster at the very centre of the morphospace (Fig. 10E, F),

**Fig. 7 | Mean cephalic shape for each geological Period compared pairwise, and pairwise centroid distances for the same.** Bottom-right half of matrix: mean cephalic shape for data grouped within each geological Period, compared in a pairwise fashion with the colour of the outline reflecting the colour of its axis. Top-left half of matrix: pairwise centroid distances for geological Periods, showing their relative positions in PCA morphospace; blue figures indicate particularly close pairwise distances (<0.100), and red figures indicate particularly far pairwise distances (>0.200). *N* = 983 independent biological samples.

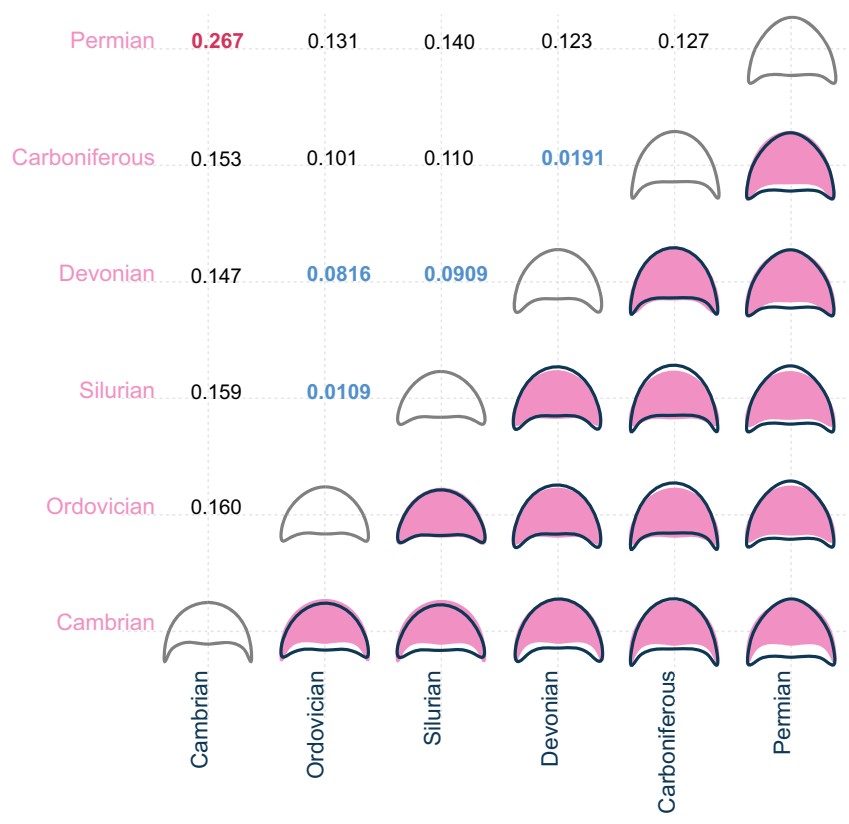

though the other clusters present again move further towards the negative PC2 area of morphospace.

## Discussion

Cephalic outline shape, alongside the presence/absence and morphology of other cephalic features such as facial sutures, eye ridges, and glabellar furrows, are key for taxonomic descriptions and assignments within Trilobita[13,61,67–73]. Indeed, many original species descriptions are based solely on cephala. Although cephalic outline shape reflects only part of the complexity of trilobite morphology with internal and post-cephalic structures also having inherent taxonomic patterns, and there is broad overlap of many orders in the central part of the morphospace (Fig. 2), a clear taxonomic signal is present in the dataset. This can most clearly be seen by the separation of Harpida from the other orders (Figs. 2–4), a result of their unique cephalon shape comprising a broad brim, high axial length, and elongate flattened genal projections—this has also been found in studies combining data from the interior and outline of the cephalon[74]. Trinucleida were also found to differ significantly from other orders in the dataset, occupying a broad area of morphospace distinct from harpids (Fig. 2) as a consequence of their thinner genal spines and differences in the outline shape of the fringe.

The relative positions of orders in our morphospace provide support for some hypotheses of high-level evolutionary relationships and historical taxonomic assignments that have been notably difficult to draw lines between. In particular, a close phylogenetic relationship between the Lichida and Odontopleurida[15,75], though other works disagree[76], and the almost identical average cephalic outline of Redlichiida and Olenida, suggesting a close relationship between the two near the origin of trilobites in morphospace[77], although only in terms of cephalic shape—the two orders differ in other important morphological aspects such as pygidium shape[78]. However, cephalic outline is clearly unable to elucidate problematic ordinal-level relationships, such as for Trinucleida, which is suggested to have its closest similarity to the unassigned, previously ptychopariid, group[79], and cephalic shape provides only one line of

evidence for taxonomic signals that may differ from those of other morphological features[61,74,77].

Distinct differences in cephalic shape between some major orders, in addition to the Harpida and Trinucleida, are also clear from the analyses. Phacopida and Proetida cluster in somewhat opposite patterns in morphospace, occupying the same area on PC1 but with Proetida more offset towards negative PC2 values (an axially longer cephalon; Fig. 2). This may reflect the differences in the orders' broad-scale evolutionary trajectories, with phacopids being more diverse earlier in the Palaeozoic and proetids later, resulting from apparent turnover events[80]. Proetida is notably constrained at the centre of the morphospace (Fig. 2), suggesting little late-Palaeozoic exploration of extreme cephalic shapes and potentially a constriction in viable cephalon forms through time (Fig. 10; though, as ever, this is may also be related to internal cephalic structures). This presumably reflects external selection pressures on proetids during the Carboniferous and Permian as the only trilobite order surviving into the Permian, over their decline to eventual extinction[15,22,76,81]. Lerosey-Aubril and Feist[23] found that proetids in the late Palaeozoic experienced many separate extinction and radiation events, though our dataset suggests this is not reflected in their cephalic shape disparity averaged across this time, and our time bins are too coarse to provide the required resolution to detect both extinctions and radiations within geological Periods. Phacopida also comprises two major clusters either side of the centre of morphospace, with the group's major differentiation being along the PC2 axis (primarily axial length of cephalon, with other cephalic margin features being reasonably consistent). Results such as this bimodal clustering suggest interesting underlying differences in morphospace occupation at lower taxonomic levels.

Redlichiida and Olenida occupy a reasonably wide area of morphospace despite the former living only during the Cambrian and the latter the Cambrian and Ordovician[78], a narrower geological range than for many other orders. The Olenida and Redlichiida may have both originated effectively contemporaneously in the early Cambrian, potentially reflecting a close evolutionary relationship, and this could explain their comparable morphospace occupation (Fig. 2) and almost identical average cephalic

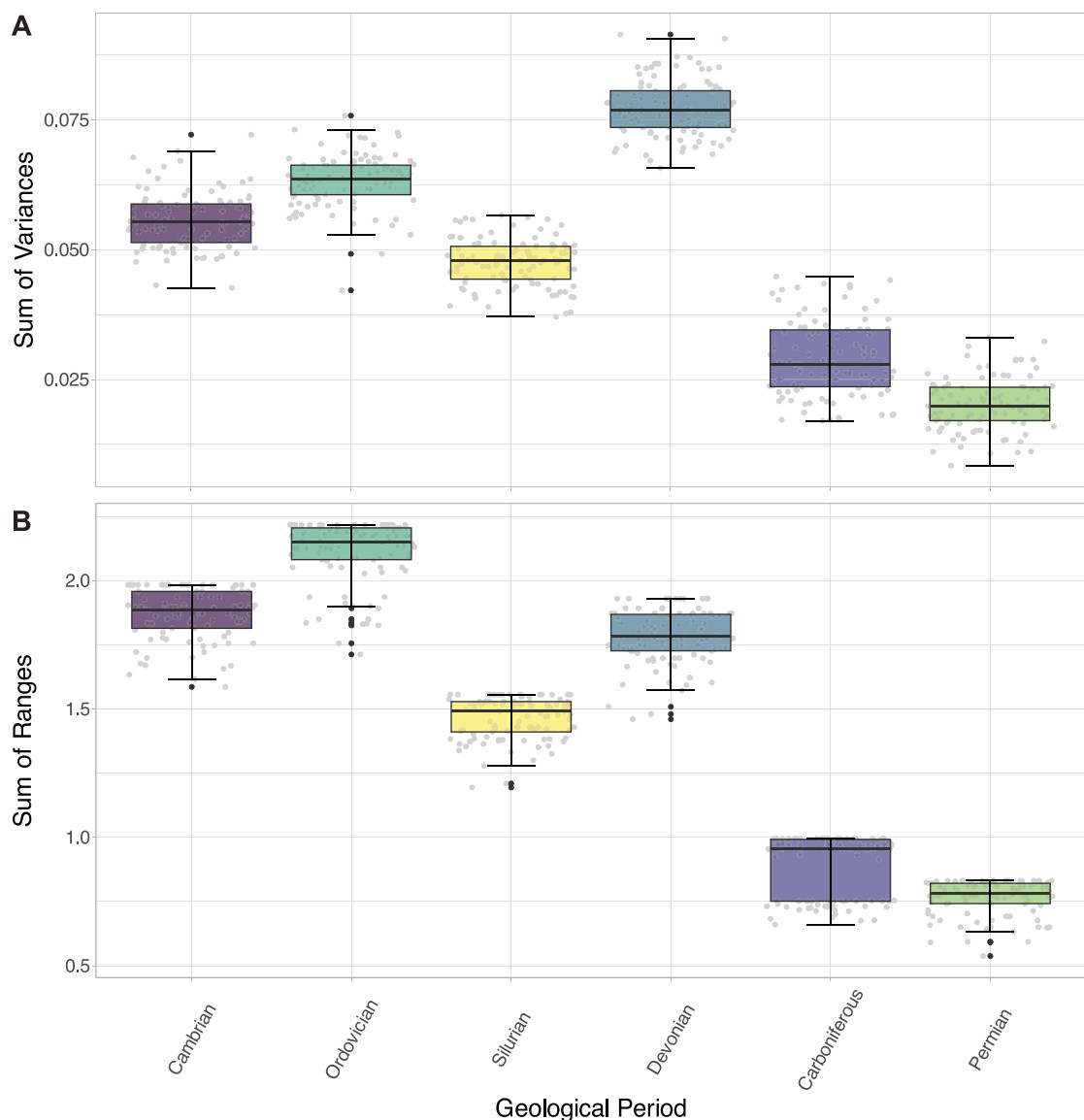

**Fig. 8 | Box and whisker plots showing disparity measures calculated for each geological Period grouping. A** sum of variances; **B** sum of ranges. The black lines show the median value, the hinges correspond to the first and third quartiles, and the error bars represent 1.5 × the interquartile range. Black plotted points represent potential outliers (outside of the error bars), and grey plotted points show the total underlying data. *N* = 983 independent biological samples.

outline shape (Fig. 4). Though the two orders differ in other morphological features not limited to the cephalon, and so shared aspects like cephalon shape could result from aspects like comparable evolutionary pressures during their contemporaneous origination. For the Redlichiida, their broad morphospace occupation might indicate the early, labile exploration of trilobites in cephalon shape during their evolution, particularly for the early Cambrian Olenellina[81], and when they might have had fewer competitors for ecological niche space that was continually partitioned[82,83], prior to constriction to a more fixed form. However, the redlichiid cephalon outline is overall centred within the morphospace, potentially disagreeing with this idea and suggesting less exploration of overall form than in orders living during the Ordovician to Devonian (Figs. 2, 6). Asaphida also occupy a relatively broad area of morphospace in both the Cambrian and Ordovician, but, as suggested by previous studies, suffered greatly in the end-Ordovician extinctions[15,84], hence their post-Ordovician disappearance in the dataset.

Some orders with lower dataset sample sizes occupy smaller areas of morphospace, such as the Aulacopleurida, Lichida, and Odontopleurida (all <40 specimens; Supplementary Table 1). However, there is not a consistent link between sampling and morphospace area occupied; several orders with

notably large sample sizes occupy smaller areas of morphospace, for example, the Asaphida and Proetida (Supplementary Table 1, Fig. 2). This likely reflects the limited cephalon outline disparity of these orders (Fig. 5), given their diversities compared to other orders. The disparity results also do not correlate well with sampling, for example, disparity measures are notably high for the Trinucleida, despite its low sample size (Supplementary Table 1, Fig. 5). Additionally, the compared disparity metrics are bootstrapped, supporting that the taxonomic trends presented here are not heavily biased by sampling, and thus likely represent real signals in the data.

However, despite the noted differences in morphospace occupation, the extensive overlapping of almost all orders (Fig. 2) indicates high similarity of cephalon outline shape within trilobites, if these taxonomic assignments are taken as reflecting true evolutionary relationships. As evidenced by the LDA results (Fig. 3), cephalon outline shape alone lacks predictive power at the order level, as only harpids can be correctly predicted with any likelihood. This is further supported by the k-means clusters, which directly interrogated the stochastic clustering of the data, without the overlain presupposed clustering enforced by taxonomy. The k-means cluster in the lower part of the space (particularly in Supplementary Fig. 1B)

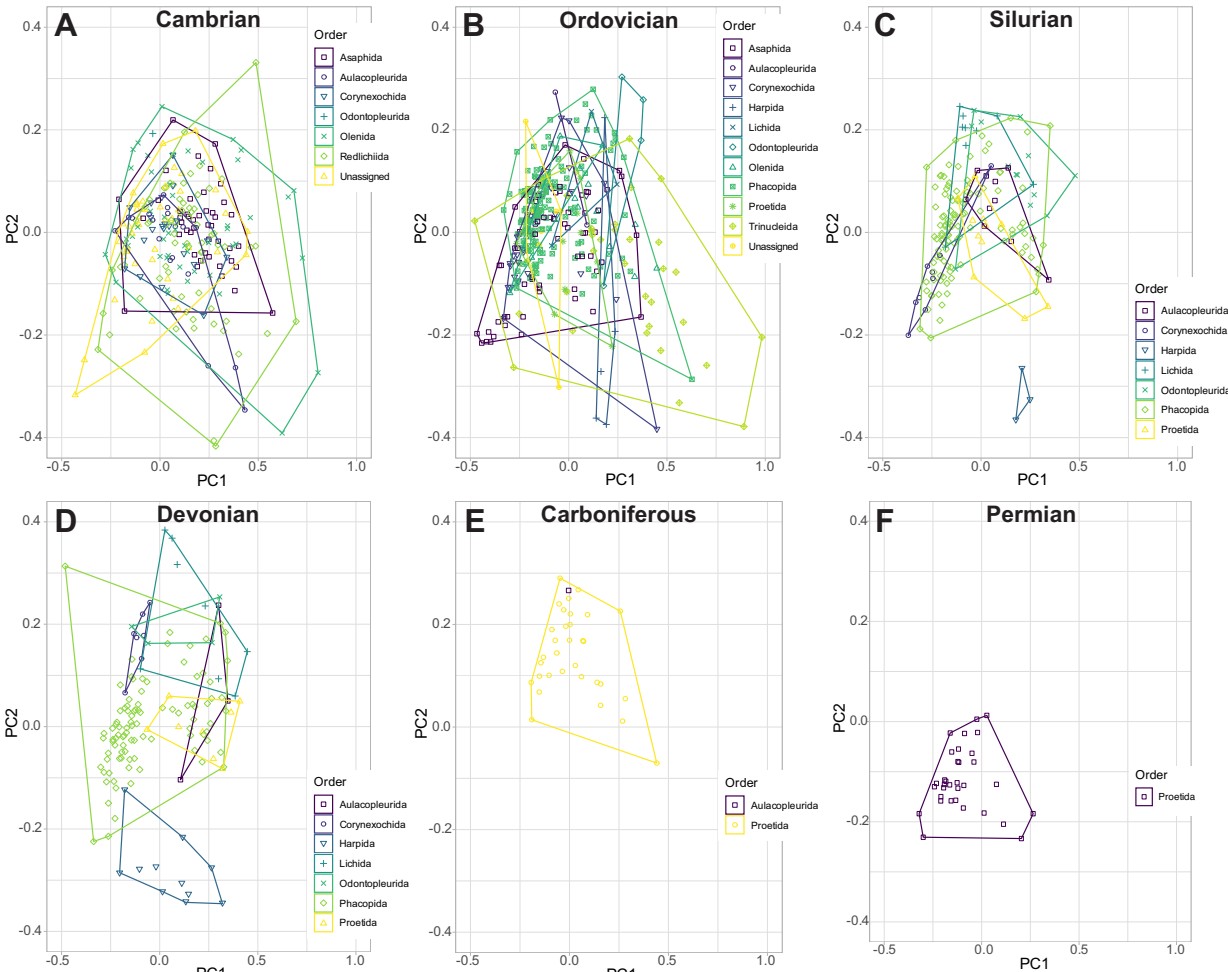

**Fig. 9 | Separate principal components analysis (PCA) morphospace plots of trilobite cephalic outlines for only specimens present in each geological Period.** The convex hulls represent the taxonomic orders present in each Period (**A–F**, labelled with the relevant Period; order denoted by colour and point shape, see legend for each of **A–F**). $N = 983$ independent biological samples.

largely reflects the location of the Harpida in morphospace (Fig. 2B), and these clusters thereby show similar average morphologies. However, despite some minor similarities, for example, the Asaphida has a similar average morphometry to the central k-means cluster, most other orders do not obviously align with any of the k-means clusters. This indicates hypothetical cephalic outline-based trilobite relationships do not generally reflect their historical taxonomic identities, at least at the order level.

Trilobite cephalic outline shape varied across the Palaeozoic, both in terms of area of occupied morphospace and average position within morphospace (Figs. 6–8). Two peaks in morphospace occupation are clear, where a peak is defined here as comprising a notable increase to a high point, immediately followed by a decrease in occupation. Overall, there is a high initial area of morphospace occupation in the Cambrian, which then increases to its maximum in the Ordovician. A subsequent fall in the Silurian is followed by a second peak in the Devonian, slightly lower than that of the Ordovician, and indeed comparable (with a slightly lower area and SoR, though higher SoV) to the high morphospace occupation of the Cambrian. Following the Devonian, morphospace occupation is much lower in the Carboniferous and Permian. An Ordovician peak in morphospace occupation differs from the previous study using cephalic outlines, where a higher disparity based on SoV (calculated in the original study) and SoR (calculated here, see Supplementary Data III) was found in the Cambrian[27]. Although, this Cambrian peak in disparity is consistent with older studies using cranidial outlines[25,26]. The second peak in morphospace occupation

during the Devonian was recovered by both sets of studies[26,27], with the Devonian SoR calculated using the Suárez and Esteve[27] data incorporated here being only just lower than for the Cambrian, and higher than for the Ordovician. The post-Devonian morphospace constriction is reflected in the k-means clustering of the dataset across the Devonian–Permian (Fig. 10); the optimal number of k-means clusters decreases following the Silurian due to the restriction of specimens to the central-lower part of the morphospace, with these fewer clusters reflecting the loss of more extreme morphologies. Overall, the results suggest a picture of changing cephalon morphometry through the Palaeozoic from axially shorter with longer genal spines, to axially longer with shorter genal spines.

Interrogation of the relative positions of centroids in the Ordovician and Devonian (during morphospace occupation peaks) compared to their previous Periods (Cambrian and Silurian respectively) suggests distinct underlying mechanisms for these expansions in morphospace area (Fig. 6A, B). During the Cambrian, cephalic shapes mostly occupy the central region of morphospace, then from the Ordovician trilobites split away from the centre along PC1 with major clusters just to the left and right of the centre (supported by the differentiation of the Cambrian and Ordovician by the LDA; Fig. 6C). These discrete clusters and differences in centroid position suggest that different trilobite cephalon shapes existed in the Cambrian compared to the post-Cambrian. This lateral shift in morphospace occupation reflects the well-established cryptogenesis problem within trilobites, where morphological similarities that link Cambrian to

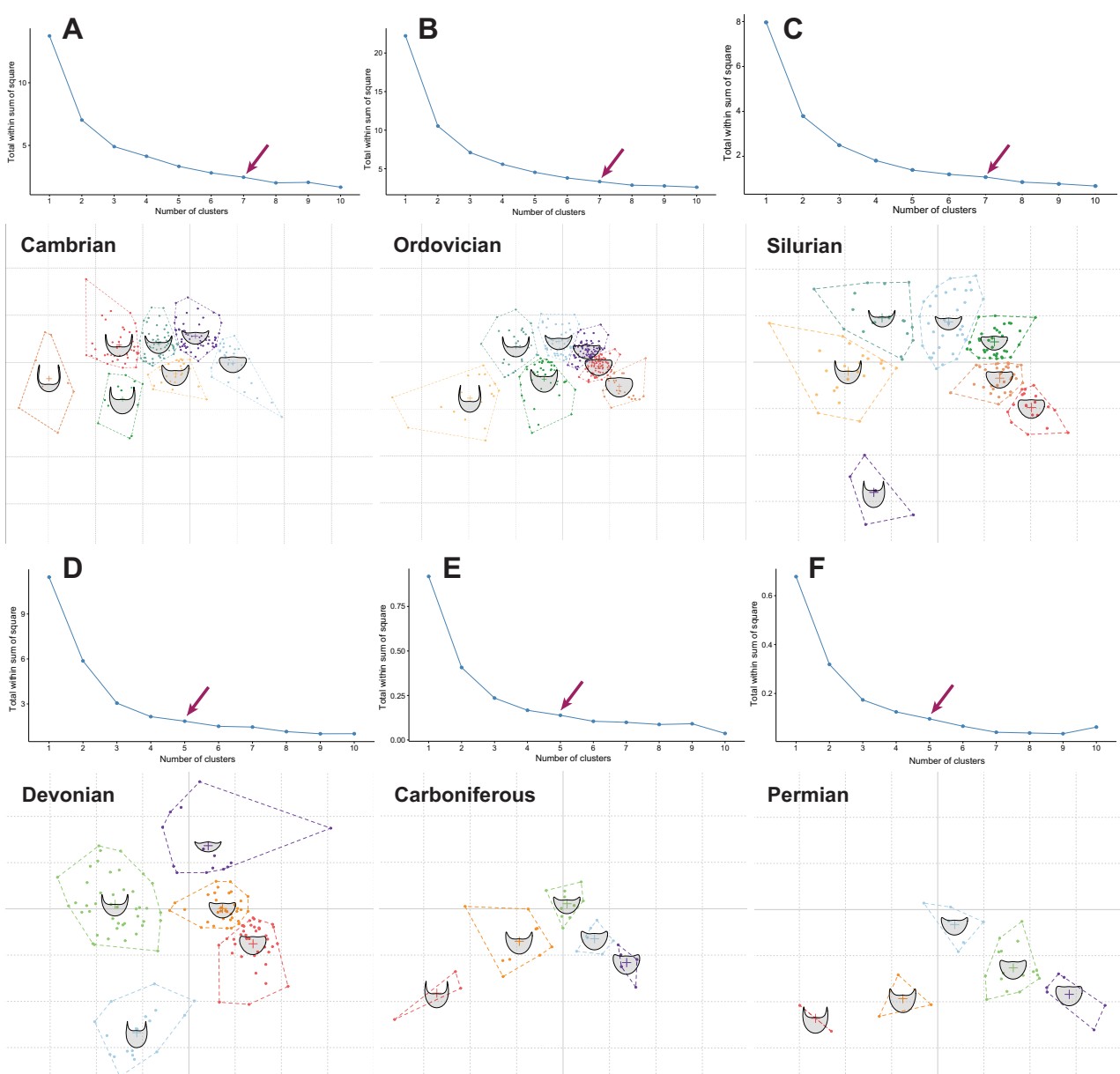

**Fig. 10 | K-means analysis displaying the natural clustering inherent in the trilobite cephalic outlines dataset, separated for each geological Period.** Both the elbow plot (displaying the most suitable number of clusters) and the clustering hypothesis are presented for each Period (**A–F**, labelled with the relevant Period). The arrows display the number of clusters at which there is an observable change (downwards trend) in *y*-axis steepness. *N* = 983 independent biological samples.

post-Cambrian trilobites have been difficult to identify (reviewed by Paterson[19]; see also Fortey[15]). Notably, the redlichiids lived only during the Cambrian, with a suborder of redlichiids, the Olenellina, having particularly different cephalic morphologies to post-Cambrian trilobites, such as a lack of facial sutures but presence of a ventral moulting suture[18,85,86]. Importantly, the Ordovician saw the origins and radiations of a range of new trilobite orders, families, and genera[14,19]. All major trilobite orders were thereby established by the late Ordovician (Fig. 9B), establishing the body plans that were embellished upon for the remainder of the Palaeozoic.

The Ordovician saw trilobites expand not just in terms of morphological differences (morphospace occupation area) but also in terms of what the 'average trilobite' looked like (morphospace centroid position). Indeed, this shift in centroid position from the Cambrian to the Ordovician is the largest of all centroid shifts in the Palaeozoic (Fig. 6), indicating that trilobites were radiating into a new set of niches at this time[74,87]. A contemporaneous taxonomic and ecological radiation is not surprising, as the

expansion of trilobites into new niches will have been facilitated by new morphological innovations, which in turn allow taxonomic groups to be diagnosed. This broadly coincides with the Great Ordovician Biodiversification Event (GOBE), later stages of the plankton revolution[88,89], and a time when trilobites explored new ecologies as juveniles and adults, with corresponding impacts on cephalic morphology. Ecological diversity in trilobites has been suggested to peak during the Ordovician[90]. For example, several groups independently evolved planktic larvae towards the late Cambrian and early Ordovician[91,92], trilobites are thought to have occupied a range of levels in the water column during the Ordovician[1,16,93–95], and some species expanded into deeper water and tropical environments[8]. The shape of the cephalon would have played an important role for moving and feeding within the water column, under hydrodynamic conditions very different to those encountered by benthic forms[93,95]. Closer to the seafloor, ecological pressures such as predation have been suggested to have led to embellishments in the trilobite exoskeleton, including the cephalon[86].

In contrast, there is no evident shift in centroid position from the Silurian to the Devonian, despite a relatively larger increase in the area of morphospace occupied and an increase in SoV and SoR across this transition (compared to the Cambrian–Ordovician; Figs. 6, 8). This expansion in morphospace but lack of centroid shift is likely driven by the relatively higher diversity of trilobites within the same orders in the Devonian compared to the Silurian (in our dataset; see also[22]), rather than broad-scale observable specialisation to new niches, though the centroid order and Period-level averaging and focus on the cephalic outline could be obscuring lower level expansions into new niches (e.g., related to regionalism, late Devonian Kellwasser event[77], etc.). Support for this interpretation comes from visual inspection of the morphospace, with expansion observable in all directions (Fig. 6B), and quantification of the morphospace occupation areas (Supplementary Table 2). Expansion in morphospace area occupation by trilobites at this time was thereby driven by morphological innovation of existing orders. This can be seen most clearly in the Harpida region of morphospace (Fig. 9C, D), with the harpids innovating following decline during the Silurian, as observed by Beech and Lamsdell[96]. Lichida and Phacopida also expand their occupation of morphospace in the Devonian, however, other trilobite orders (e.g., Aulacopleurida, Corynexochida, Proetida) did not contribute to this increase in morphospace occupation (Fig. 9C, D), demonstrating heterogeneity in response.

Constrictions in morphospace occupation (Ordovician to Silurian; Devonian to Carboniferous; Carboniferous to Permian) are also not associated with large shifts in centroid position in the dataset (Fig. 6). This indicates that reduction in morphospace occupation is associated with declining diversity of trilobites (Fig. 9), but not driving a lateral shift in morphospace occupation[12]. The severity of the decrease in SoR and SoV from the Ordovician to Silurian (*c.* 30%) and Devonian to Carboniferous (*c.* 50–60%) and from the Carboniferous to Permian (*c.* 20–25%) indicates some selectivity for removal of taxa towards the extremes of the morphospace[12]. For the Devonian to Carboniferous reduction, this is likely related to the extinction of harpids, lichids and odontopleurids, which occupy the extremes of the Devonian morphospace, with only aulacopleurids and proetids surviving, which are positioned close to the Carboniferous centroid. For the Ordovician to Silurian and Carboniferous to Permian, reductions in the morphospace area are not related to the removals of whole orders, but instead lead to a reduction in the intra- and inter-order disparities (Fig. 9).

Previous studies have linked selectivity for extinction at the end-Ordovician to the ecology of trilobite larvae[84], while global eustatic perturbations, anoxia of bottom waters, and broad environmental changes are associated with Devonian extinction events, which likely led to the extinction of deep-water groups[23]. While we have not quantified juvenile ecologies, and so cannot test their correlation with adult cephalic shape, nor have we quantified specific internal cephalic structures, such as eyes or facial sutures, and we have sampled only at a very coarse temporal resolution, it is notable that some evidence for extinction selectivity is returned from these analyses. Further work incorporating the larval ecology of trilobites in this dataset[84,92], details from the interior of the cephalon (e.g., using the scheme of Serra et al.[60]), and a higher temporal resolution will allow explicit testing of selectivity for particular larval ecologies at the end Ordovician, as well as the impact of eye reduction and facial suture shape as a response to global sea level changes in the end-Devonian[97,98].

Following the end-Devonian extinction, proetids expanded into areas of morphospace they had not previously occupied (Fig. 9D–F), including a dense band close to an area originally dominated by Phacopida and a space proximal to the positive extreme of PC2 previously occupied by Harpida. This increase in disparity reflects an increase in generic level diversity that is decoupled from the stagnant order level diversity. In turn, this demonstrates the capacity that proetids had to adapt following the Devonian mass extinctions[23], potentially facilitated by the extinction of harpids and phacopids. This proetid expansion was likely driven by high origination rates in the early Carboniferous, however, this origination rate dropped dramatically in the Viséan (*c.* 350 Ma) and did not recover[23]. The continued

winnowing of trilobite diversity, and, with it, disparity, was likely the result of major environmental changes associated with the icehouse Carboniferous and subsequent formation of Pangaea[23]. Further global environmental changes occurred in the Permian, and their impact on trilobite diversity and disparity and are reflected by the further constriction of morphospace area occupation at the Carboniferous–Permian boundary, and eventual extinction of trilobites.

Measures of morphospace occupation can be influenced by sample size, and this is potentially problematic for quantitative comparisons of morphospace area later in the Palaeozoic, with the two Periods with the lowest morphospace areas occupied and lowest disparities being those with the smallest sample sizes (Fig. 6, Supplementary Table 2). However, as for orders, morphospace area occupied does not consistently track with sample size, for example, sample size is higher in the Silurian than the Devonian, but morphospace area occupied is higher for the Devonian (Supplementary Table 2). Further, the sampling accurately reflects the much lower species richness present in the Carboniferous and Permian resulting from the prior extinctions of almost all trilobite orders[15,22,23].

Despite these evident trends in cephalic outline disparity and trilobite diversity, the predictive power of cephalon morphometry for geological Period is generally poor (Fig. 6D), as for the taxonomic orders. Only the Cambrian and Ordovician groupings were predictive, with the average Ordovician cephalon being axially longer, and the Cambrian cephalon having longer genal spines (Fig. 7). Thus, this large and broadly sampled dataset suggests that cephalon outlines could not generally be used to predict the order assignment or the occupied geological Period of an unknown trilobite.

In conclusion, a large dataset of almost 1000 2D trilobite cephalon outlines was used to explore the extent of trilobite occupation of morphospace, and to determine whether this cephalic morphometry was associated with order assignment or geological Period occupied. The data show significant differences in morphospace occupation and disparity, supporting a reasonably strong taxonomic signal and changes in disparity through the Palaeozoic. Two peaks in trilobite cephalic shape disparity are observed, attributed to the Ordovician and the Devonian. Importantly, the expansion of morphospace occupation in the Ordovician was likely due to the occupation by trilobites into new niches (linked to the origin of all trilobite orders without Cambrian origins), contrasting it to the second peak of morphospace occupation in the Devonian, which was likely linked solely to within-order taxonomic diversification. However, due to extensive overlapping of morphospace occupation, clustering analyses indicate that the order or Period assignment of a new unknown trilobite would be almost impossible to predict with any accuracy based on 2D cephalic outline morphometry. Only the cephalic outlines of Harpida, and of Cambrian and Ordovician trilobites, are predictive.

## Methods
### Using the cephalon outline as a proxy for trilobite disparity
A large part of trilobite diversity and disparity of forms is manifested in their cephalic morphometries; the shapes and sizes of the cephala of different groups. The variation in functional morphology of the cephalon is intrinsically linked to the varied life modes of trilobites; different cephalic shapes and structures have been suggested to be adaptations to different feeding modes[99–104], life modes[16,18,64,105], or specific behaviours[27,85,106]. However, the evolution and extent of disparity of the trilobite cephalon remains unclear, with uncertainty around the unstable high-level trilobite taxonomy[19,76,107], the potential homology of cephalic structures[73,108,109], and the adaptation of cephalic shape to hypothetical life mode.

While cephalic outline cannot capture changing morphology of structures such as the facial sutures or glabella[61], the lack of homologous morphological points otherwise shared between all trilobite groups makes this a way to compare across all trilobites in our dataset. Other schemes, combining landmarks and semilandmarks, also facilitate this, by coding landmarks inapplicable for certain groups as missing[60]. Furthermore, while 3D data might be expected to be more representative of form, previous

studies comparing the use of 2D and 3D trilobite morphometric data on comparable scales suggest 2D data is sufficient for evolutionary hypothesis-testing, as it reconstructs the same patterns as 3D data[33]. Compaction and flattening of specimens preserved in shales does increase the variance of landmark positions, alongside abaxial and anterior splaying of genal regions[38], however, not on the same scale as interspecific differences in cephalic shape. Though, it is important that this cephalic shape data is considered in conjunction with disparity in other morphological aspects of the trilobite form, particularly internal cephalic features that are ecologically important (e.g., the eyes, see Thomas[110]) and can individually differ[61].

### Dataset construction

Cephalic outlines of 386 specimens (published by Suárez and Esteve[27], original dataset of 400 specimens but with agnostids removed), were added to a new dataset collected from photographs of 597 trilobite cephala representing at least 520 species total. These were gathered from the descriptive literature, online museum databases accessed through iDigBio, and additional museum photography by the authors (for references, search terms, and dates of access, see Supplementary Data I and II). Specimens were photographed from the following museum collections: the Senckenberg Museum (183 specimens, representing 122 species), Sedgwick Museum, Cambridge (62 specimens, representing 46 species), Staatliches Museum für Naturkunde Karlsruhe (10 specimens, representing 9 species), and Natural History Museum, London (112 specimens, representing 81 species). Only adult specimens were included because cephalon shape can dramatically change across development, with even whole features (e.g., genal spines, marginal spines, etc.) variously appearing and disappearing[33,111,112]. Only reasonably intact cephala, without extreme deformation and with at least one librigena in place, were included; singular missing librigenae were reconstructed based on the shape and location of the articulated librigena. Work by Webster and Hughes[38] suggested that flattening has a consistent effect on the morphometric data representing different species, and so that flattening deformation would likely have a minor impact on broad-scale results.

Cephalic outline semilandmark data for specimens were obtained using tpsDIG2[113]. For all specimens two curves were created: (1) from the tip of the left genal spine (or, in the absence of a genal spine, the posterior-left margin of the cephalon) clockwise around the anterior of the cephalon to the tip of the right genal spine (or, in the absence of a genal spine, the posterior-right margin of the cephalon); (2) from the tip of the right genal spine (or in the absence of a genal spine, the posterior-right margin of the cephalon) around the posterior of the cephalon to the tip of the left genal spine (or, in the absence of a genal spine, the posterior-left margin of the cephalon). Each curve consisted of 64 evenly spaced semilandmarks, giving a total cephalic outline for each specimen constructed of 128 semilandmarks (see Supplementary Fig. 2). The data published by Suárez and Esteve[27] were resampled to give an equivalent number of outline semilandmarks for all specimens. Metadata for all species included order assignment following Adrain[107] and geological Period occupied following Jell and Adrain[114]. Species traditionally included within the 'Ptychopariida' were grouped as 'Unassigned' due to the problematic nature of the clade[15,27,107]. See Supplementary Tables 1 and 2 for a summary of the dataset composition.

### Statistics and reproducibility

Geometric morphometric analyses and multivariate statistical analyses were performed to test the following null hypotheses, using the dataset of 983 trilobite specimens described above:

1. Trilobite taxonomic orders show no difference in their cephalic outline morphometries.
2. There were no broad-scale changes in cephalon morphometry through geological time.

The data gathering and analytical protocol is presented in Supplementary Fig. 2 and described below.

Firstly, a total dataset morphospace was produced using elliptical Fourier analysis (EFA) with 13 harmonics that represent 99.9% of the variation retained. Principal Components Analyses (PCA) were then used to visualise relevant patterns of shape variation and analyse differences in morphospace occupation between groupings, including taxonomic groups and geological age groups. The PCA groupings were further interrogated by calculating the convex hull areas and comparing the centroid distances between groups. MANOVA tests were performed to test for differences in the distributions of principal components (PC) scores between groupings, and subsequent pairwise tests used to interrogate the revealed PC score differences (alpha level of 0.05 with Bonferroni correction = 0.05/n pairings). Only PCs 1 and 2 were plotted and used for the subsequent analyses and interrogations presented here; the morphospace scree plots (Fig. 2D) show that these are responsible for 62.5% and 24% of the variation respectively. PC3 represents 4.2% of the morphospace variation; on visualising the outline changes across PC3 it is clear this variation results from sample noise, likely due to minor specimen compression and distortion, and so PCs from 3 onwards were discarded.

Linear Discriminant Analyses (LDA) were also performed to visualise morphospace occupation when differences were maximised between the groupings. LDA cross-validation tables were produced to provide an alternative method of exploring the LDA overlap of groupings. These cross-validation tables inform on the probability that a new data point within each group would be incidentally placed within each other group, and therefore give an estimation of the predictiveness of cephalic outline for each grouping in the dataset.

K-means clustering analyses were performed to explore how cephalon morphometric data stochastically formed groups in EFA morphospace. Mean cephalon outline shapes were produced for the groupings to visualise any differences in cephalic morphometry across taxonomic groups or through time. Finally, disparity measures were calculated and plotted to compare disparity between the different groups, including the sums of variances (SoV; spread of specimens in morphospace, or how tightly packed they are with the occupied area) and sums of ranges (SoR; breadth of morphospace occupation, more sensitive to outliers)[115,116]. Multiple disparity metrics were calculated as methodological studies have found this to be key to extracting all pertinent information[12,115]. Pairwise t-tests (alpha level of 0.05 with Bonferroni correction = 0.05/n pairings) were then performed to test for differences in the measured disparities of the groups.

All analyses were run and plots created in RStudio[117,118] version 2023.09.1 + 494, using the following packages: momocs for outline geometric morphometric analyses and related statistics[119]; ggplot2 for plotting[120]; dispRity to calculate disparity measures[121]. All data, including metadata (specimen accession information, taxonomy, geological age), raw cephalon outline landmark coordinates, and coordinates for PCs 1 and 2, are available in Supplementary Data I and online[39] at https://osf.io/vz9a5/. All R code, including a script for data preparation and for all analyses performed, are available online at https://osf.io/vz9a5/.

### Reporting summary

Further information on research design is available in the Nature Portfolio Reporting Summary linked to this article.

### Data availability

Source data underlying graphs and charts are available in Supplementary Data 1 and open access[39] at https://osf.io/vz9a5/ or https://doi.org/10.17605/OSF.IO/VZ9A5 (the latter including the photographs used to produce the cephalic outlines).

### Code availability

All R code needed to run the analyses are also available open access at https://osf.io/vz9a5/ or https://doi.org/10.17605/OSF.IO/VZ9A5. All analyses were run and plots created in RStudio[117,118] version 2023.09.1 + 494, using the following packages: momocs for outline geometric morphometric

analyses and related statistics[119]; ggplot2 for plotting[120]; dispRity to calculate disparity measures[121].

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

## Acknowledgements

We thank two anonymous reviewers for their constructive and detailed reviews that improved the manuscript. We are grateful to Jana Bruthansová (National Museum Prague, Prague), Richard Howard and Zoe Hughes (Natural History Museum UK, London), Matt Riley (Sedgwick Museum, Cambridge), Mónica M. Solórzano Kraemer (Senckenberg Museum, Frankfurt), and Julien Kimmig (Staatliches Museum für Naturkunde Karlsruhe) for providing access to specimens at their respective institutions. In addition, we thank all the museums, volunteers and staff who uploaded images of trilobites to iDigBio, which facilitated data collection for this research during periods of remote working during early 2021. HBD was funded under a Swiss National Science Foundation Sinergia grant (198691). SP was supported for this work by a Herchel Smith Postdoctoral Fellowship (University of Cambridge) and NERC IRF NE/X017745/1 (University of Exeter).

## Author contributions

Harriet B. Drage: conceptualisation, methodology, software, validation, formal analysis, investigation, data curation, writing—original draft, writing—review & editing, visualisation, project administration. Stephen Pates: conceptualisation, methodology, software, validation, formal analysis, investigation, data curation, writing—original draft, writing—review & editing, project administration.

## Competing interests

The authors declare no competing interests.
