## [Peer review file · Communications Biology]

Distinct causes underlie double-peaked trilobite morphological disparity in cephalic shape

Corresponding Author: Dr Harriet Drage

Version 0:

Reviewer comments:

Reviewer #1

(Remarks to the Author)
Comments

I am not a native English speaker and do not make comments about language in the paper.

1. In this manuscript, the analyses of morphospace for trilobites are based on their cephalic outlines rather than cephalic structures. Although the cephalic outlines of trilobites are likely to be homologous or convergence, this study would be certain meaningful to understand macroevolutionary of trilobites during the Palaeozoic. The authors should emphasize 'cephalic outline or shape' in this paper, for example, in the title, 'Distinct causes underlie double-peaked trilobite morphological disparity on cephalic shape' should be better.
2. In methods and materials, the morphological changes of the same species during ontogeny should be considered and explained. For example, the genal spine appears in juvenile specimens of asaphids, while it possibly disappears in adult individuals. The presence or absence of the genal spine would change the cephalic outline of trilobites.
3. As the authors noted, 'An Ordovician peak in morphospace occupation differs from the previous study using cephalic outlines, where a peak in occupation was found in the Cambrian (Suárez and Esteve, 2021), though is consistent with older studies using cranidial outlines (Foote, 1991, 1993b)'. I believe that it would be very meaningful to add the comparison and discussion of morphospace occupation based on different methods (e.g., cephalic outline, cranidial outline and others).

Detailed points. Those in the text listed by the line and page numbers on the word.

Line 17, Page 1, '983 cephalon outlines'---'983 cephalon outlines of c. 520 species'

Line 21, Page 1, 'Cephalic disparity peaks in the Ordovician and Devonian'---'Cephalic shape disparity peaks in the Ordovician and Devonian'

Line 23, Page 1, 'late Ordovician'---'Late Ordovician'

Line 59, Page 3, 'early Cambrian'---'Cambrian Stage 3 (Series 2)'

Line 61, Page 3, 'diversity'---'species diversity'

Line 144, Page 6, '2.1 Using the cephalon as a proxy for trilobite disparity'---'2.1 Using the cephalic outline as a proxy for trilobite disparity'

Lines 278-281, Page 11, what is '(ax.)'? , it should be 'sag.'?

Lines 683-685, Page 27, pages?

Line 701, Page 28, Historical Biology 36, 473-484.

Lines 976-979, Pages 40-41, pages?

Figure 3A and C, the texts of trilobite orders are overlapping.

Reviewer #2

(Remarks to the Author)

First of all I really appreciated reading this manuscript, I think that the work raises very interesting ideas that are undoubtedly a great contribution to the knowledge of the evolution and ecology of this fossil group. At the same time, the attempt to cover through a morphological approach a group as diverse and morphologically complex as trilobites makes it of great interest for trilobite workers and paleontologists in general.

Here the authors explore trilobite disparity through time, considering the whole range of the Clade, and analyze the presence of a taxonomic signal in morphospace. Further, they test whether both order-level taxonomy and/or the Period of

occurrence can be predicted according to the shape of the outline of the cephalon. To do this, the authors generated a vast database obtained from previous publication and their own work. Regarding the latter, the authors must be recognized for the effort this implies.

In general, the manuscript is well organized and contains all necessary information. The methodology is well explained, the amount of analysis that was carried out gives robustness to the results and their interpretations. Tables and illustrations are informative on the results, and the references appear to be relevant.

My main concern is related to the scope of the database in the attempt to answer these questions and the conclusions reached regarding the evolutionary relationships of trilobites. It is widely proven by various authors that the structures or substructures of an organism provide different information regarding disparity patterns (eg. Oudot et al. 2019, Cole and Hopkins 2020, Bault et al. 2022, Holmes 2023, among others). Particularly the outline of the cephalon not only takes information about a single structure of the trilobite body, but also omits information regarding the internal features, all of which are key for taxonomic assignment and for ecological or functional interpretations. For example, information about facial suture (involved in molting), size, shape, and position of the eye, inflation of the glabella (associated with feeding habits), are lost. On the other hand, the outline of the cephalon alone seems to be a rather homogeneous structure throughout the evolutionary history of trilobites to be used as the single variable to predict high level taxonomy or Period occurrence.

For example:

Lines 439 – 441: “the almost identical average cephalic outline of Redlichiida and Olenida, suggesting a close relationship between the two near the origin of trilobites in morphospace”

Lines 467 – 470: “The Olenida and Redlichiida may have both originated effectively contemporaneously in the early Cambrian, potentially reflecting a close evolutionary relationship, and this could explain their comparable morphospace occupation (Fig. 3) and almost identical average cephalic outline shape (Fig. 5).”

Making these inferences based solely on the outline of the cephalon can be misleading. In particular these two clades may result in similar cephalic outlines, as shown in this manuscript, but the overall morphology is quite different (see the thorax of Redlichiida for instance or even the absence of facial suture or tiny pygidium).

Further, a recent paper (Holmes 2023) that analyzes the disparity of the cephalic outline and internal features (glabellar and eye ridge morphology) found contrasting patterns for these structures, with the latter increasing more gradually, in line with the increase in taxonomic diversity. Also, the main cephalic shape differences between the orders analyzed in Bault et al. 2022 (<https://doi.org/10.1111/pala.12623>) are related to the position of the facial suture and the relative width of the frontal lobe (features that are left out by the data acquisition protocol in this manuscript). In this sense, I believe that phylogenetic conclusions are somewhat ambitious regarding the database used and the discussion related to the taxonomic signal and evolutionary relationships between orders should be led with care. I suggest that the section “4.1 Taxonomic signal in cephalon outline morphometry” needs revision keeping in mind that only the contour is being analyzed.

However, the amount of analysis carried out on this large database provides a lot of important information that can be complemented with the information provided by other authors who considered other structures and data acquisition techniques, which were mentioned in the manuscript (eg. Foote, Hopkins 2014, Vargas-Parra and Hopkins 2022, Oudot et al 2019, Suárez and Esteve 2021, Bault et al. 2022, Serra et al. 2023), and contribute from that place to the discussion.

Comments in relation to the discussion in the section “4.2 Two peaks in morphospace occupation during the Palaeozoic”:

*Lines 509 – 512: Overall, there is a high initial area of morphospace occupation in the Cambrian, which then increases to its maximum in the Ordovician. A subsequent fall in the Silurian is followed by a second peak in the Devonian, slightly lower than that of the Ordovician.

SoR (and area of Convex Hull) for the Devonian is actually slightly lower than that of the Cambrian, according to the results presented here, there would be one peak in the Ordovician and two moments of high morphospace occupation (high SoR) during the Cambrian and Devonian.

It seems that in the Devonian there is a more disperse occupation of morphospace (peaking SoV) although its range of occupation remains lower than that of the Ordovician and equal to the Cambrian.

*Lines 513 – 515: An Ordovician peak in morphospace occupation differs from the previous study using cephalic outlines, where a peak in occupation was found in the Cambrian (Suárez and Esteve, 2021),

This Cambrian peak (Suarez and Esteve, 2021) is based on comparisons of sum of variances only, so this would be comparable with the increase in SoV documented in the Ordovician and the clear Devonian peak in your dataset.

*Lines 568 - 574: This expansion in morphospace is likely driven by the relatively higher diversity of trilobites within the same orders in the Devonian compared to the Silurian (in our dataset; see also Bault et al., 2022), rather than specialisation to new niches

Why don't these innovations represent exploration into new niches? The expansion in the morphospace through the occupation of the extremes may reflect new strategies. In the Silurian - Devonian transition there is a marked turnover of trilobite fauna (Bault et al. 2022), and the Devonian is characterized by the coexistence of at least 3 of them, which may lead

to high disparity in this period. For example, the Devonian evolutionary fauna mainly developed following major environmental changes, increasing their occupation towards outer platform and slope. The Kellwasser Fauna is characterized by important morphological changes such as the eye-reduction.

*Lines 581 – 584: Constrictions in morphospace occupation (Ordovician to Silurian; Devonian to Carboniferous; Carboniferous to Permian) are also not associated with large shifts in centroid position in the dataset (Fig. 7). This indicates that reduction in morphospace occupation is associated with declining diversity of trilobites (Fig. 10), but no clear extinction specificity can be observed.

In the Ordovician – Silurian and Devonian – Carboniferous transitions there is a decrease in the SoR, but also a significant decrease in the SoV suggesting a selection towards the elimination of the margins.

*Line 568: Fig. 7

Add Figure 9. PCA plots are mostly considered for visualization, when analyzing morphospace occupation is better to look at the measured indices. The increase in the SoV and SoR from Silurian to Devonian suggest inflation of morphospace by occupation of the margins.

Other minor comments relate to:

Methodology:

*Lines 157 – 159: “While cephalic outline does not capture potential disparity across the whole cephalon (Holmes, 2023), the lack of homologous morphological points otherwise shared between all trilobite groups makes this a way to utilise the broadest taxonomic dataset.”

May be eliminate or rewrite this sentence. There is a recent paper published in Scientific Data that discusses this and provides a landmark template for Trilobita, capturing cephalon and pygidium morphology using a geometric morphometric approach. <https://doi.org/10.1038/s41597-023-02724-9>

Results:

*Lines 260 – 261: Proetida show clustering in an opposing area of morphospace to the Phacopida, with the majority of specimens falling between the two major phacopid clusters and the rest found along more negative PC2 values.

It does not seem to be in opposite sides, rather it seems that Proetida is “included” in Phacopida’s morphospace. Also by LDA cross-validation the entry of a new specimen of Proetida is more probable to be predicted as Phacopida (0.64) than Proetida itself (0.27).

Discussion:

Lines 439 – 441: I think you ment to cite: Bault et al. 2022 Palaeontology <https://doi.org/10.1111/pala.12623>

Line 509: Do you mean Figs 7 – 9?

Version 1:

Reviewer comments:

Reviewer #1

(Remarks to the Author)

I have received the manuscript author revised. Some questions have been replied and explained. So, I consider that it can be accepted now.

Reviewer #2

(Remarks to the Author)

Very briefly, the authors explore trilobite disparity through time, considering the whole range of the Clade, and analyze the presence of a taxonomic signal in morphospace. Further, they test whether both order-level taxonomy and/or the Period of occurrence can be predicted according to the shape of the outline of the cephalon.

As mentioned in my previous review, the manuscript is well organized and contains all necessary information. The methodology is well explained, the amount of analysis that was carried out gives robustness to the results and their interpretations. Tables and illustrations are informative on the results, and the references appear to be relevant.

All my concerns and observations were properly addressed by the authors, I have no further suggestions to make. The importance of the work carried out by Drs. Drage and Pates has already been pointed out in my previous review, I think this is a valuable contribution to the field of palaeontology, particularly regarding macroevolutionary and macroecological dynamics assessed by novel approaches.

v

Responses to reviewer comments

Reviewer #1:

Thanks very much to reviewer 1 for providing detailed, helpful feedback on our manuscript. We greatly appreciate the time the reviewer took in reading and commenting on the manuscript, and their assistance in aiding us to improve this study!

Comments

I am not a native English speaker and do not make comments about language in the paper.

1. In this manuscript, the analyses of morphospace for trilobites are based on their cephalic outlines rather than cephalic structures. Although the cephalic outlines of trilobites are likely to be homologous or convergence, this study would be certain meaningful to understand macroevolutionary of trilobites during the Palaeozoic. The authors should emphasize 'cephalic outline or shape' in this paper, for example, in the title, 'Distinct causes underlie double-peaked trilobite morphological disparity on cephalic shape' should be better.

Thanks for your summary of the paper, and noting this aspect – we have ensured that outline or shape are noted when referring to the cephalic morphometry results throughout the manuscript, and changed the title. We have also added text in several places throughout to draw more attention to the importance of considering this outline data in conjunction with those from other structures.

2. In methods and materials, the morphological changes of the same species during ontogeny should be considered and explained. For example, the genal spine appears in juvenile specimens of asaphids, while it possibly disappears in adult individuals. The presence or absence of the genal spine would change the cephalic outline of trilobites.

This is a good point, thanks to the reviewer for noting this. We have clarified in the methods that we included only adult specimens in the dataset for this reason.

3. As the authors noted, 'An Ordovician peak in morphospace occupation differs from the previous study using cephalic outlines, where a peak in occupation was found in the Cambrian (Suárez and Esteve, 2021), though is consistent with older studies using cranial outlines (Foote, 1991, 1993b)'. I believe that it would be very meaningful to add the comparison and discussion of morphospace occupation based on different methods (e.g., cephalic outline, cranial outline and others).

We have added a more detailed discussion of the findings based on these different methods at the noted point.

Detailed points. Those in the text listed by the line and page numbers on the word.

Line 17, Page 1, '983 cephalon outlines'---'983 cephalon outlines of c. 520 species'

Line 21, Page 1, 'Cephalic disparity peaks in the Ordovician and Devonian'---

'Cephalic shape disparity peaks in the Ordovician and Devonian'

Line 23, Page 1, 'late Ordovician'---'Late Ordovician'

Line 59, Page 3, 'early Cambrian'---'Cambrian Stage 3 (Series 2)'

Line 61, Page 3, 'diversity'---'species diversity'

Line 144, Page 6, '2.1 Using the cephalon as a proxy for trilobite disparity'---

'2.1 Using the cephalic outline as a proxy for trilobite disparity'

Lines 278-281, Page 11, what is '(ax.)?', it should be 'sag.'?

Lines 683-685, Page 27, pages?

Line 701, Page 28, Historical Biology 36, 473-484.

Lines 976-979, Pages 40-41, pages?

Figure 3A and C, the texts of trilobite orders are overlapping.

All suggested changes have been made, except for the below for the stated reasons:

- 'late Ordovician' is correct because 'late' is a descriptor, not a formally named Epoch in the ICS timescale (the corresponding formal name is 'Lower Ordovician')
- we prefer to keep the phrase 'Trilobites originated during the early Cambrian' rather than specifying Stage 3 because, while the earliest trilobite body fossils are found close to/ at the base of Stage 3, it is plausible that trilobites first originated prior to this (consistent with trace fossils), which would place their origins in the Terreneuvian and not Stage 3 (e.g., see Daley et al. 2018 *PNAS*; Paterson et al. 2019 *PNAS*)
- we agree that it is not ideal that the order names overlap in Figure 3, however, the positions of the names are actually informative because they show the position of the group centroid. We have added this into the figure caption to clarify on this point.

Reviewer #2 (Remarks to the Author):

Thanks very much to reviewer 1 for their kind words on our manuscript. We greatly appreciate the time the reviewer took in reading and commenting on the manuscript, and believe that their suggestions helped improve this study!

First of all I really appreciated reading this manuscript, I think that the work raises very interesting ideas that are undoubtedly a great contribution to the knowledge of the evolution and ecology of this fossil group. At the same time, the attempt to cover through a morphological approach a group as diverse and morphologically complex as trilobites makes it of great interest for trilobite workers and paleontologists in general.

Here the authors explore trilobite disparity through time, considering the whole range of the Clade, and analyze the presence of a taxonomic signal in morphospace. Further, they test whether both order-level taxonomy and/or the Period of occurrence can be predicted according to the shape of the outline of the cephalon. To do this, the authors generated a vast database obtained from previous publication and their own work. Regarding the latter, the authors must be recognized for the effort this implies.

In general, the manuscript is well organized and contains all necessary information. The methodology is well explained, the amount of analysis that was carried out gives robustness to the results and their interpretations. Tables and illustrations are informative on the results, and the references appear to be relevant.

My main concern is related to the scope of the database in the attempt to answer these questions and the conclusions reached regarding the evolutionary relationships of trilobites. It is widely proven by various authors that the structures or substructures of an organism provide different information regarding disparity patterns (eg. Oudot et al. 2019, Cole and Hopkins 2020, Bault et al. 2022, Holmes 2023, among others). Particularly the outline of the cephalon not only takes information about a single structure of the trilobite body, but also omits information regarding the internal features, all of which are key for taxonomic assignment and for

ecological or functional interpretations. For example, information about facial suture (involved in molting), size, shape, and position of the eye, inflation of the glabella (associated with feeding habits), are lost. On the other hand, the outline of the cephalon alone seems to be a rather homogeneous structure throughout the evolutionary history of trilobites to be used as the single variable to predict high level taxonomy or Period occurrence.

Thanks for your comment regarding using just the cephalic outline data in this study, we absolutely agree that internal structures are key for taxonomic assignment and evolutionary interpretations. We noted this in the original submission in regard to our decision to use this data (section 2.1), and discussed these other aspects in the discussion. However, this is indeed a really important point, so we have added more text drawing attention to this throughout and at the relevant areas suggested in the specific comments below.

For example:

Lines 439 – 441: “the almost identical average cephalic outline of Redlichiida and Olenida, suggesting a close relationship between the two near the origin of trilobites in morphospace”

Lines 467 – 470: “The Olenida and Redlichiida may have both originated effectively contemporaneously in the early Cambrian, potentially reflecting a close evolutionary relationship, and this could explain their comparable morphospace occupation (Fig. 3) and almost identical average cephalic outline shape (Fig. 5).”

Making these inferences based solely on the outline of the cephalon can be misleading. In particular these two clades may result in similar cephalic outlines, as shown in this manuscript, but the overall morphology is quite different (see the thorax of Redlichiida for instance or even the absence of facial suture or tiny pygidium).

That is true, their overall morphology can be quite different, and we only intended to provide a line of reasoning based on this one data source – we have caveated this in the text at the relevant places to ensure these are not taken as hard inferences and to note other possibilities. However, our original text refers to the more-recently-elevated order Olenida (previously ptychopariids, which has facial sutures, etc.), not the Olenellidae (the latter lacking facial sutures and having a more different morphology, but being currently included within order Redlichiida).

Further, a recent paper (Holmes 2023) that analyzes the disparity of the cephalic outline and internal features (glabellar and eye ridge morphology) found contrasting patterns for these structures, with the latter increasing more gradually, in line with the increase in taxonomic diversity. Also, the main cephalic shape differences between the orders analyzed in Bault et al. 2022 (<https://doi.org/10.1111/pala.12623>) are related to the position of the facial suture and the relative width of the frontal lobe (features that are left out by the data acquisition protocol in this manuscript). In this sense, I believe that phylogenetic conclusions are somewhat ambitious regarding the database used and the discussion related to the taxonomic signal and evolutionary relationships between orders should be led with care. I suggest that the section “4.1 Taxonomic signal in cephalon outline morphometry” needs revision keeping in mind that only the contour is being analyzed.

We agree, and do not intend to make concrete statements regarding evolutionary relationships based on this data, only to make suggestions that might augment future

studies. For this reason, we have altered the text throughout this section to clarify on these points and ensure any suggestions based on the data are not put forward too strongly. The first paragraph of this section also directly notes the limitations of this dataset, as we do not wish to obfuscate this.

Additionally, we do agree regarding the facial suture, though we are actually considering this structure as part of a separate study because it is integral to a separate analysis of behaviour that distracted from the results presented in this manuscript!

However, the amount of analysis carried out on this large database provides a lot of important information that can be complemented with the information provided by other authors who considered other structures and data acquisition techniques, which were mentioned in the manuscript (eg. Foote, Hopkins 2014, Vargas-Parra and Hopkins 2022, Oudot et al 2019, Suárez and Esteve 2021, Bault et al. 2022, Serra et al. 2023), and contribute from that place to the discussion.

Thank you for the kind words on our study, indeed we hope this study, taken with the others mentioned, serve to contribute to our knowledge on trilobite evolution!

Comments in relation to the discussion in the section “4.2 Two peaks in morphospace occupation during the Palaeozoic”:

*Lines 509 – 512: Overall, there is a high initial area of morphospace occupation in the Cambrian, which then increases to its maximum in the Ordovician. A subsequent fall in the Silurian is followed by a second peak in the Devonian, slightly lower than that of the Ordovician.

SoR (and area of Convex Hull) for the Devonian is actually slightly lower than that of the Cambrian, according to the results presented here, there would be one peak in the Ordovician and two moments of high morphospace occupation (high SoR) during the Cambrian and Devonian.

It seems that in the Devonian there is a more disperse occupation of morphospace (peaking SoV) although its range of occupation remains lower than that of the Ordovician and equal to the Cambrian.

This is true – when referring to ‘two peaks’ we had been considering these as periods of high morphospace occupation following a notable increase from the previous Period, which is why we discussed these as being in the Ordovician and Devonian. You are right, however, that the Cambrian does show a comparably high occupation of morphospace to the Devonian. The text in this section doesn’t contradict this, so this hasn’t been altered, though we have added text noting that the Cambrian occupation is also particularly high and comparing it to the Devonian. Further, we have defined our usage of the term ‘peaks’ at the beginning of this section to ensure this is clear.

*Lines 513 – 515: An Ordovician peak in morphospace occupation differs from the previous study using cephalic outlines, where a peak in occupation was found in the Cambrian (Suárez and Esteve, 2021),

This Cambrian peak (Suarez and Esteve, 2021) is based on comparisons of sum of variances only, so this would be comparable with the increase in SoV documented in the Ordovician and the clear Devonian peak in your dataset.

This has been clarified in the text, thanks. We have also calculated the SoRs for the Suárez and Esteve (2021) data (those data from that publication used in this manuscript), which also show a peak in the Cambrian (slightly higher than the values for the Ordovician and Devonian). These results are included in Appendix III, and explicitly noted at this point in the discussion.

*Lines 568 - 574: This expansion in morphospace is likely driven by the relatively higher diversity of trilobites within the same orders in the Devonian compared to the Silurian (in our dataset; see also Bault et al., 2022), rather than specialisation to new niches

Why don't these innovations represent exploration into new niches? The expansion in the morphospace through the occupation of the extremes may reflect new strategies. In the Silurian - Devonian transition there is a marked turnover of trilobite fauna (Bault et al. 2022), and the Devonian is characterized by the coexistence of at least 3 of them, which may lead to high disparity in this period. For example, the Devonian evolutionary fauna mainly developed following major environmental changes, increasing their occupation towards outer platform and slope. The Kellwasser Fauna is characterized by important morphological changes such as the eye-reduction.

This is a good point – we have clarified at this point in the text, and added text noting that this finding and focus on the cephalic outline could be obscuring lower-level patterns such as those noted by the reviewer.

*Lines 581 – 584: Constrictions in morphospace occupation (Ordovician to Silurian; Devonian to Carboniferous; Carboniferous to Permian) are also not associated with large shifts in centroid position in the dataset (Fig. 7). This indicates that reduction in morphospace occupation is associated with declining diversity of trilobites (Fig. 10), but no clear extinction specificity can be observed.

In the Ordovician – Silurian and Devonian – Carboniferous transitions there is a decrease in the SoR, but also a significant decrease in the SoV suggesting a selection towards the elimination of the margins.

We thank the reviewer for highlighting this, and we have altered the two relevant paragraphs of the discussion to reflect evidence for extinction selectivity towards the margins of our space.

*Line 568: Fig. 7

Add Figure 9. PCA plots are mostly considered for visualization, when analyzing morphospace occupation is better to look at the measured indices. The increase in the SoV and SoR from Silurian to Devonian suggest inflation of morphospace by occupation of the margins.

Done

Other minor comments relate to:

Methodology:

*Lines 157 – 159: “While cephalic outline does not capture potential disparity across the whole cephalon (Holmes, 2023), the lack of homologous morphological points

otherwise shared between all trilobite groups makes this a way to utilise the broadest taxonomic dataset.”

May be eliminate or rewrite this sentence. There is a recent paper published in Scientific Data that discusses this and provides a landmark template for Trilobita, capturing cephalon and pygidium morphology using a geometric morphometric approach. <https://doi.org/10.1038/s41597-023-02724-9>

It is true that this is discussed in the given reference - we have added an additional note regarding this paper in the identified location. It is a shame that this work was well underway at the time this paper was published, as it is a good paper and future collaborations would certainly be productive!

Results:

*Lines 260 – 261: Proetida show clustering in an opposing area of morphospace to the Phacopida, with the majority of specimens falling between the two major phacopid clusters and the rest found along more negative PC2 values.

It does not seem to be in opposite sides, rather it seems that Proetida is “included” in Phacopida’s morphospace. Also by LDA cross-validation the entry of a new specimen of Proetida is more probable to be predicted as Phacopida (0.64) than Proetida itself (0.27).

It is true that Proetida mostly falls within the Phacopida convex hull, though we were referring to the actual clusters of specimens within morphospace rather than the limits of the convex hulls – we have clarified this point and the overlapping of the convex hulls in the text.

Discussion:

Lines 439 – 441: I think you ment to cite: Bault et al. 2022 Palaeontology
<https://doi.org/10.1111/pala.12623>

Done

Line 509: Do you mean Figs 7 – 9?

Done

Thanks again to both reviewers and the editors for their work on this submission. We greatly appreciate the assistance in improving our manuscript!

Best wishes,
Dr Harriet B. Drage & Dr Stephen Pates